



# Invisible aerosol layers: improved lidar detection capabilities by means of laser-induced aerosol fluorescence

Benedikt Gast[1], Cristofer Jimenez[1], Albert Ansmann[1], Moritz Haarig[1], Ronny Engelmann[1],
Felix Fritzsch[1], Athena A. Floutsi[1], Hannes Griesche[1], Kevin Ohneiser[1], Julian Hofer[1], Martin Radenz[1],
Holger Baars[1], Patric Seifert[1], and Ulla Wandinger[1]

[1]Leibniz Institute for Tropospheric Research, Leipzig, Germany

**Correspondence:** Benedikt Gast (bgast@tropos.de)

**Abstract.** One of the most powerful instruments for studying aerosol particles and their interactions with the environment is atmospheric lidar. In recent years, fluorescence lidar has emerged as a useful tool for identifying aerosol particles due to its link with biological content. Since 2022, this technique has been implemented in Leipzig, Germany. This paper describes the experimental setup and data analysis, with a special emphasis on the characterization of the new fluorescence channel centered at 466 nm. The new capabilities of the fluorescence lidar are examined and corroborated through several case studies. Most of the measurement cases considered are from the spring and summer of 2023, when large amounts of biomass-burning aerosol from the huge forest fires in Canada were transported to Europe. The fluorescence of the observed aerosol layers is characterized. For wildfire smoke, the fluorescence capacity was typically in the range of $2$–$7 \times 10^{-4}$, which aligns well with the values reported in the literature, with slightly larger values. The key aspects of this study are the capabilities of the fluorescence lidar technique, which can potentially improve not only the typing but even the detection of aerosol particles. In several measurement cases with an apparently low aerosol load, the fluorescence channel clearly revealed the presence of aerosol layers that were not detectable with the traditional elastic-backscatter channels. This capability is discussed in detail and linked to the fact that fluorescence backscattering is related to aerosol particles only. A second potential of the fluorescence technique is the distinction between non-activated aerosol particles and hydrometeors, given water's inability to exhibit fluorescence. A smoke-cirrus case study suggests an influence of the aerosol layer on cloud formation, as it seems to affect the elastic backscatter coefficient within the cloud passing time. These mentioned applications promise huge advancements towards a more detailed view of the aerosol-cloud interaction problem.

## 1 Introduction

A crucial player in the atmospheric system are aerosol particles, given their role in various processes that ultimately shape the Earth's energy and hydrological budgets. Firstly, aerosol particles scatter and absorb radiation, affecting the energy balance on a global scale. By serving as cloud condensation nuclei (CCN) or ice nucleating particles (INPs), these particles can impact the microphysical properties of water clouds (Liu et al., 2014), making them more or less reflective depending on the aerosol situation (Twomey, 1974, 1977; Twomey et al., 1984). Because the microphysical properties of a cloud play a major role in





its development and the formation of precipitation, aerosol conditions can further affect the extension and lifetime of cloud
events and therefore the global albedo (Albrecht, 1989; Stevens and Feingold, 2009). Highly absorbing aerosol particles might
even affect clouds via the so-called semi-direct effect, which can manifest, e.g., in the evaporation of cloud droplets due to
an aerosol-heated environment (Hansen et al., 1997). Aerosol effects on the ice phase of cloud formation only complicate the
picture. Multiple efforts have been made to analyze the role of aerosol particles as INPs in mixed-phase clouds via heteroge-
neous freezing and the global effect (Lohmann, 2017). As for cirrus clouds, recent studies suggest that heterogeneous freezing
in cirrus clouds, particularly via smoke particles, needs to be explored in more detail (Ansmann et al., 2021; Veselovskii et al.,
2022a; Mamouri et al., 2023; Ansmann et al., 2024a, b). To improve our understanding of these complex aerosol-cloud inter-
action processes, reliable detection and characterization of atmospheric aerosol particles are essential.

Multi-wavelength polarization lidars are powerful tools to detect and characterize aerosol particles. After decades of study,
several classification schemes are available in the literature (Floutsi et al., 2023; Groß et al., 2013; Burton et al., 2012), mostly
relying on intensive (i.e., concentration-independent) optical properties such as the lidar ratio, particle depolarization ratio and
Ångström exponent. However, although clear signatures can be expected for some particle types (e.g., low particle depolariza-
tion and low lidar ratios for marine aerosol particles), some limitations remain. Distinguishing between stratospheric smoke
and volcanic sulfates or separating between tropospheric smoke and urban pollution remain difficult tasks, as their typical
ranges of values for particle depolarization and lidar ratios partially overlap. Additional information, such as the fluorescence
of atmospheric aerosol particles, may be required to address these typing difficulties (Veselovskii et al., 2022b).

Laser-induced fluorescence is a known process and several remote-sensing applications are based on it. Fluorescence lidars
have been around for a while, but their application has mostly focused on water composition (Palmer et al., 2013; Cadondon
et al., 2020) and vegetation (Edner et al., 1994). In the atmosphere, efforts have mostly gone towards the detection of single
molecules (Mcllrath, 1980; Wang et al., 2021). To investigate the fluorescence of atmospheric aerosol particles, the experi-
ments have been mostly performed through in situ probing (Pinnick et al., 2004; Pan, 2015; Zhang et al., 2019; Kawana et al.,
2021). A first hint towards the observation of atmospheric aerosol fluorescence with lidar came in 2005, when Immler et al.
(2005) observed an inelastic backscatter signal in the water vapor Raman channel (i.e., at 407 nm) that was not produced by
Raman scattering. They attributed it to the laser-induced fluorescence emission from biomass-burning (BB) aerosol particles
and already linked the aerosol fluorescence to organic compounds. Pan (2015) analyzed the fluorescence of aerosol particles
by measurements with an ultraviolet aerodynamic particle sizer (UV-APS) and reported a spectral range for atmospheric fluo-
rescence of 400 to 650 nm, when excited at 355 nm. Later, the first atmospheric fluorescence lidars were set up, but most were
small and dedicated to fluorescence measurements only.

Rao et al. (2018) and Li et al. (2019) used Nd:YAG lasers at 266 nm and 355 nm, respectively, and studied the backscattered
light in one elastic-backscatter (at the excitation wavelength) and one fluorescence channel. Both instruments looked only at the
boundary layer. Saito et al. (2018) studied the fluorescence of atmospheric pollen with a lidar at 355 nm excitation wavelength
and found a similar spectral range of 400 to 600 nm for the fluorescence emission. The first advanced multi-channel atmo-
spheric lidar system with fluorescence technology was implemented at Lindenberg observatory of the German Meteorological
Service (DWD). For the first time, Reichardt et al. (2018) used a spectrometer to measure the fluorescence of aerosol particles





in the middle and upper troposphere. They characterized the fluorescence of mineral dust and BB aerosol and underlined the capabilities of fluorescence measurements to study aerosol-cloud coexistence by enabling the observation of aerosol particles even inside clouds. Veselovskii et al. (2020, 2021) extended the concept to a practical approach, in which only a single broadband fluorescence channel was added into a multi-wavelength lidar system at Lille, France. They also described a retrieval scheme for the computation of the fluorescence backscatter coefficient out of the signal ratio between the fluorescence and the nitrogen channels. Their observations also confirmed the potential of the fluorescence lidar technique to study aerosol particles inside clouds (Veselovskii et al., 2022a). Veselovskii et al. (2022b) showed that fluorescence measurements can improve the aerosol classification with lidar. They proposed, for the first time, a simple classification scheme that combines the linear depolarization ratio with the fluorescence capacity (Reichardt, 2014), which is defined as the ratio of the fluorescence backscatter coefficient to an elastic particle backscatter coefficient (e.g., at 532 nm as used in this study). With this approach, they were able to discriminate between smoke, mineral dust, pollen and urban aerosol, as pollen and smoke showed significantly higher values of fluorescence capacity than urban aerosol and mineral dust. Reichardt et al. (2023) described a procedure for absolute calibration of spectrometric fluorescence measurements and proposed a method to correct for the systematic fluorescence error in water vapor measurements with Raman lidar, which is significant for dry and strongly fluorescent aerosol layers. They also emphasized that the spectrum's shape is closely related to the aerosol type. Veselovskii et al. (2023) presented an approach to measure rough fluorescence spectra with a lidar with five discrete broadband fluorescence channels at Moscow, Russia. They reported advancements in aerosol typing with this approach compared to a single fluorescence channel. Smoke and urban aerosol particles could be discriminated even at high relative humidity and in the presence of hygroscopic growth.

In this work, we explore the observational capabilities of an atmospheric fluorescence lidar utilizing measurements performed in Leipzig, Germany, with an upgraded system since 2022. A detailed description of the new experimental setup is provided in Sec. 2. The analysis of several measurement case studies is presented in Sec. 3. Our findings corroborate the results obtained by previous studies on the capabilities of fluorescence lidars and deepen the discussion in the field of aerosol studies utilizing fluorescence lidar observations. We discuss a unique new capability that is special to this measurement approach. Because it is sensitive to particles only, a fluorescence channel can potentially improve not only the typing but even the detection of aerosol particles. Sec. 3.2.3 provides an in-depth analysis of the reasons for this increased sensitivity of the fluorescence channel to aerosol particles. An exceptional smoke-cirrus interaction case presented in Sec. 3.2.4 highlights the importance of the ability to detect thin aerosol layers in the upper troposphere and lower stratosphere (UTLS) region for the investigation of cirrus cloud formation. Furthermore, it corroborates and expands the initial work on the detection of fluorescence signals inside ice clouds. The paper concludes in Sec. 4.

## 2 Experimental Setup

### 2.1 Implementation of a fluorescence channel in the MARTHA lidar system

The Multi-wavelength Atmospheric Raman lidar for Temperature, Humidity and Aerosol profiling (MARTHA) is a lab-based lidar system at the Leibniz Institute for Tropospheric Research (TROPOS) in Leipzig. It emits electromagnetic radiation at



three wavelengths (355 nm, 532 nm and 1064 nm) with an overall pulse energy of about 1.2 J at a repetition rate of 30 Hz and collects the backscattered radiation with a large main mirror, which measures 80 cm in diameter. A detailed description of the MARTHA system is given in Mattis et al. (2002), Schmidt et al. (2013) and Jimenez et al. (2019).

To measure the laser-induced fluorescence of atmospheric aerosol, the MARTHA lidar system was upgraded by adding a discrete fluorescence channel into the receiving unit in 2022. To facilitate comparability, the spectral range of the channel was set in the same wavelength range as in Veselovskii et al. (2020). A 44 nm-wide interference filter from Alluxa centered at 466 nm is used to select a part of the fluorescence spectrum of fluorescing aerosol particles. Because of the similar features, a first comparison of the results obtained in Lille, France, and Leipzig, Germany, is possible.

The backscattered fluorescence intensity depends on the aerosol situation, but in general, it is much weaker than elastic

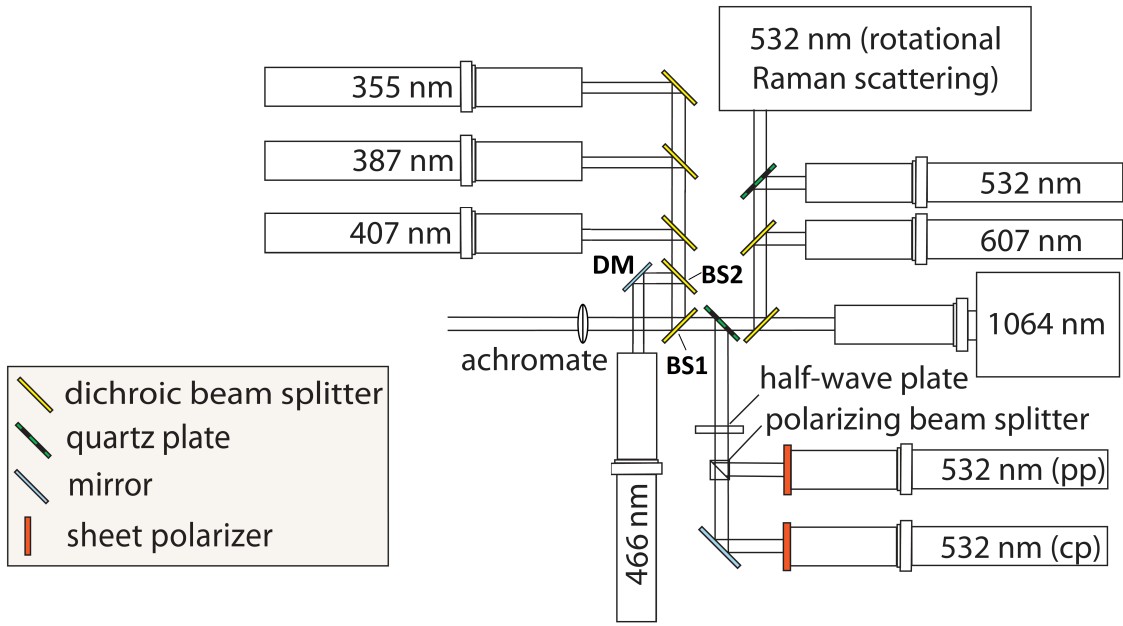

**Figure 1.** Setup of the far-range receiving unit of the MARTHA system after implementing the fluorescence channel (graphic adapted from Schmidt (2014)). DM: dielectric mirror; BS: beam splitter.

backscatter signals. Therefore, the signal-to-noise ratio in the new channel must be as high as possible. The features that made the MARTHA system suitable to detect fluorescence are its large telescope area and the high-power laser. The second and third harmonic generation setups allowed an increase in the laser energy at 355 nm, sending 6 ns pulses with an energy of about 350 mJ. The new setup of the MARTHA far-range (FR) receiver, including the fluorescence channel, is displayed in Fig.

1. The new detection channel was placed in the branch with the lower wavelengths. Therefore, the first long-pass beam splitter (BS1) was replaced to ensure the complete reflection of the intended fluorescence spectral band. A second beam splitter (BS2) was added. It transmits the shorter wavelengths to the elastic-backscatter and Raman channels related to the UV laser emission at 355 nm and reflects the longer wavelengths towards the fluorescence channel. As the fluorescence signal can be 4–5 orders



of magnitude weaker than elastic backscattering (Veselovskii et al., 2020), sufficient suppression of the elastic returns in the new channel was essential to measure fluorescence. The two new beam splitters received customized coatings from Laseroptik GmbH to guarantee high suppression of the elastic-backscatter (and Raman) lines and only a little loss of fluorescence return. Two interference filters were placed in tandem to suppress the elastic components further.

## 2.2 Analytical scheme of the fluorescence backscatter coefficient

The lidar system was operated manually and only if no rain was expected. Complete nights were collected since 2022 and were analyzed with a focus on the fluorescing properties of the observed aerosol layers. A second important step is the derivation of the new products. The procedure is described as follows: The aerosol fluorescence backscatter coefficient was obtained by forming the ratio of the fluorescence ($\boldsymbol{P}_\mathrm{F}$) to the nitrogen Raman ($\boldsymbol{P}_\mathrm{R}$) signal, similarly as in Veselovskii et al. (2020). Both signals can be described in terms of the lidar equation:

$$\boldsymbol{P}_\mathrm{F} = \beta_\mathrm{F} \boldsymbol{T}_\mathrm{F} C_\mathrm{F} \tag{1}$$

$$\boldsymbol{P}_\mathrm{R} = \beta_\mathrm{R} \boldsymbol{T}_\mathrm{R} C_\mathrm{R}. \tag{2}$$

$\boldsymbol{T}_\mathrm{R}$ and $\boldsymbol{T}_\mathrm{F}$ denote the atmospheric transmission at the Raman and fluorescence wavelength ranges, respectively, and $C_\mathrm{R}$ and $C_\mathrm{F}$ the corresponding lidar calibration constants. By dividing Eq. (1) by Eq. (2), the following expression for the aerosol fluorescence backscatter coefficient $\boldsymbol{\beta}_\mathrm{F}$ can be derived:

$$\boldsymbol{\beta}_\mathrm{F} = \frac{\boldsymbol{P}_\mathrm{F}}{\boldsymbol{P}_\mathrm{R}} \frac{\boldsymbol{T}_\mathrm{R}}{\boldsymbol{T}_\mathrm{F}} \frac{C_\mathrm{R}}{C_\mathrm{F}} \boldsymbol{\beta}_\mathrm{R}. \tag{3}$$

The Raman backscattering $\boldsymbol{\beta}_\mathrm{R}$ is computed using the following expression in terms of the Rayleigh molecular backscatter coefficient ($\boldsymbol{\beta}_\mathrm{mol}$):

$$\boldsymbol{\beta}_\mathrm{R} = D_\mathrm{R} \boldsymbol{N}_\mathrm{N_2} = 0.78 D_\mathrm{R} \boldsymbol{N}_\mathrm{mol} = 0.78 \frac{D_\mathrm{R}}{D_\mathrm{mol}} \boldsymbol{\beta}_\mathrm{mol}, \tag{4}$$

with $\boldsymbol{N}_\mathrm{N_2}$ and $\boldsymbol{N}_\mathrm{mol}$ the number density of nitrogen and air molecules, respectively. $D_\mathrm{R}/D_\mathrm{mol}$ accounts for the Raman to Rayleigh backscatter differential-cross-section ratio. These cross sections were determined theoretically using Eqs. (20) and (14) in Adam (2009), resulting in theoretical values of $D_\mathrm{R}^* = 2.7344 \times 10^{-34}\,\mathrm{m^2\,sr^{-1}}$ and $D_\mathrm{mol} = 3.10875 \times 10^{-31}\,\mathrm{m^2\,sr^{-1}}$.

## 2.3 Technical considerations for the calibration of the fluorescence channel

To derive the particle fluorescence backscatter coefficient from Eq. (3), the traditional method, using a particle-free reference height, cannot be applied, due to the unknown fluorescence response of the background aerosol. Instead, a characterization of the channel's system efficiencies is needed. The contribution of each component in the respective detection path was carefully determined to infer the overall efficiencies and build the lidar-constant ratio $C_\mathrm{R}/C_\mathrm{F}$.

The first point to consider is the bandwidth of the interference filters. For the 387 nm nitrogen Raman channel, with a bandwidth of 2.7 nm, only 95 % of the theoretical Raman cross section can reach the detector, reducing the actual cross section



**Table 1.** Transmittances $\mathcal{T}$ or reflectances $\mathcal{R}$ of all optical elements in the $387\,\mathrm{nm}$ and $466\,\mathrm{nm}$ FR channels. PMT quantum efficiencies, PMT gain ratio and overall ratio of the lidar calibration constants in both channels.

| FR channel | Nitrogen Raman (387 nm) | Fluorescence (466 nm) |
|---|---|---|
| Common beam splitter (BS2) | $\mathcal{T}_1 = 97.1\,\%$ | $\mathcal{R}_1 = 98\,\%$ |
| Further (unique) optical elements | beam splitter 407 nm: $\mathcal{T}_2 = 94.5\,\%$ beam splitter 387 nm: $\mathcal{R}_2 = 95\,\%$ | dielectric mirror (DM) $\mathcal{R}_3 = 99.75\,\%$ |
| Interference filters | $\mathcal{T}_3 = 70\,\%$ | $\mathcal{T}_4 = 92.5\,\%$ ($\mathcal{T}_4{}^2$ because of 2 filters) |
| Neutral-density filters (example) | $\mathrm{OD} = 1.3$ $\mathcal{T}_{\mathrm{ND}} \approx 0.0213$ | no neutral-density filters $\mathcal{T}_{\mathrm{ND}} = 1$ |
| Product $\prod_{i=1}^{N} \mathcal{T}_i \prod_{j=1}^{M} \mathcal{R}_j$ | $\mathcal{R}_2\,\mathcal{T}_1\,\mathcal{T}_2\,\mathcal{T}_3\,\mathcal{T}_{\mathrm{ND}} = 0.01299$ | $\mathcal{R}_1\,\mathcal{R}_3\,\mathcal{T}_4{}^2 = 0.8364$ |
| PMT quantum efficiency PMT gain ratio | 34.66 % $\eta_{\mathrm{gain,R}}/\eta_{\mathrm{gain,F}} = 1.4155$ | 25.13 % |
| Overall ratio of the lidar calibration constants | $C_{\mathrm{R}}/C_{\mathrm{F}} \approx 0.0303$ | |

at the detector to $D_{\mathrm{R}} = 0.95 \times D_{\mathrm{R}}^* = 2.59768 \times 10^{-34}\,\mathrm{m^2\,sr^{-1}}$. This value is then used in Eq. (4) together with the molecular backscatter computed based on the temperature and pressure profiles provided by the Global Data Assimilation System
(GDAS) (Rodell et al., 2004) to derive the Raman backscatter coefficient. As the laser power, pulse length, and telescope area are the same for both detection channels, the lidar-constant ratio $C_{\mathrm{R}}/C_{\mathrm{F}}$ simplifies to the ratio of the channel efficiencies. This ratio comprises the transmittances or reflectances of the optical elements (such as beam splitters, mirrors, interference filters and neutral-density filters) and the detection efficiencies of the detectors. The transmittances and reflectances of the optical components are collected in Tab. 1. As the neutral-density (ND) filters are eventually changed depending on the atmo-
spheric and system conditions, only one exemplary set of ND filters, which is representative of most of the cases studied in this manuscript, was chosen for Tab. 1. When determining the ND filter transmission, the spectral dependence provided by the manufacturer (Thorlabs) was considered. The detection efficiencies of the photomultiplier tubes (PMTs) are split into electrical gain and detector efficiency. The ratio of the electrical gains was obtained by swapping the detectors in the nitrogen Raman and fluorescence channels and building the ratio of the mean signals measured by both detectors for each channel. This test
yielded a PMT gain ratio ($\eta_{\mathrm{gain,R}}/\eta_{\mathrm{gain,F}}$) of 1.4155, indicating a higher gain of the nitrogen Raman channel's PMT. As for the



detector surface, the so-called quantum efficiency accounts for the amount of photoelectrons generated by the cathode divided by the number of incident photons. This efficiency depends on the photon wavelength (Wright, 2017). The spectrally resolved quantum-efficiency data provided by Hamamatsu were considered to assess the PMT type used in the MARTHA system. The maximum efficiency of the detectors is about 35 %, and the values at the wavelength ranges of the two lidar channels were

determined by interpolation from the provided data points and averaging over the filter width of the interference filter in the fluorescence channel. This resulted in values of $\eta_{\mathrm{qe,R}} = 34.66\,\%$ and $\eta_{\mathrm{qe,F}} = 25.13\,\%$ for the quantum efficiencies of the used PMT type in the nitrogen Raman (386–388 nm) and fluorescence channel (444–488 nm), respectively, and we finally obtained a ratio of $\eta_{\mathrm{qe,R}}/\eta_{\mathrm{qe,F}} = 1.379$.

After these considerations, the ratio of the lidar calibration constants can be calculated from the efficiency ratios of the optical

elements, detector gain and spectral response as follows:

$$\frac{C_{\mathrm{R}}}{C_{\mathrm{F}}} = \frac{\mathcal{R}_2\,\mathcal{T}_1\,\mathcal{T}_2\,\mathcal{T}_3\,\mathcal{T}_{\mathrm{ND}}}{\mathcal{R}_1\,\mathcal{R}_3\,\mathcal{T}_4{}^2}\,\frac{\eta_{\mathrm{gain,R}}}{\eta_{\mathrm{gain,F}}}\,\frac{\eta_{\mathrm{qe,R}}}{\eta_{\mathrm{qe,F}}}. \tag{5}$$

For the set of ND filters considered in Tab. 1 (OD 1.3 in the nitrogen Raman and no ND filters in the fluorescence channel) it results in a value of $C_{\mathrm{R}}/C_{\mathrm{F}} \approx 0.0303$.

The remaining unknown in Eq. (3) is the ratio $\boldsymbol{T}_{\mathrm{R}}/\boldsymbol{T}_{\mathrm{F}}$ of the atmospheric transmissions (ground to target) at the Raman and

fluorescence wavelengths, respectively. The molecular part $(\boldsymbol{T}_{\mathrm{R}}/\boldsymbol{T}_{\mathrm{F}}|_{\mathrm{mol}})$ is calculated straightforwardly from the extinction and backscatter coefficients; the aerosol contribution to the transmission ratio $(\boldsymbol{T}_{\mathrm{R}}/\boldsymbol{T}_{\mathrm{F}}|_{\mathrm{par}})$ requires previous knowledge about the aerosol backscatter coefficient. For the profile-based analysis, the aerosol optical properties are determined with the traditional Raman technique (Ansmann et al., 1990, 1992). The particle backscatter coefficient at high temporal resolution was obtained via a constant-based approach, in which a previous profile-based retrieval is needed to calculate the lidar constants,

which are then used to compute high-resolution products out of the elastic-backscatter and Raman signals (Baars et al., 2017). In general, the particle atmospheric transmission differs little at the two wavelengths, making the effect on the fluorescence backscatter coefficient small, partially because only about 80 nm separate the central wavelengths of the Raman and fluorescence channels. For the cases with low and medium aerosol loads (see Sec. 3.2), the error in case of non-consideration of the differential transmission was in the range of 2–6 %. In case of an unusually high aerosol optical depth, like on the 4 July 2023

(see Sec. 3.1), the error was in the order of 10 %. Thus, the differential particle transmission at the two wavelengths was considered to guarantee the quality of the fluorescence backscatter coefficient, even above strongly backscattering aerosol layers. But still, the assumption of an appropriate Ångström exponent is necessary, which imposes an uncertainty of $\pm 1$–7 % on the determined $\boldsymbol{T}_{\mathrm{R}}/\boldsymbol{T}_{\mathrm{F}}|_{\mathrm{par}}$, depending on the optical thickness of the present aerosol layers.

The data set acquired in Leipzig was then analyzed in a semi-automatic manner, setting the calibration constants and the

reference height (particle-free) manually for each case. The fluorescence capacity $\boldsymbol{G}_{\mathrm{F}}$,

$$\boldsymbol{G}_{\mathrm{F}} = \frac{\boldsymbol{\beta}_{\mathrm{F}}}{\boldsymbol{\beta}_{532}}, \tag{6}$$

was calculated as the ratio of the fluorescence backscatter coefficient $(\boldsymbol{\beta}_{\mathrm{F}})$ to the elastic particle backscatter coefficient at 532 nm $(\boldsymbol{\beta}_{532})$. Furthermore, data from the collocated portable Raman lidar Polly$^{\mathrm{XT}}$ (Engelmann et al., 2016) at TROPOS were used to provide quality-assured depolarization profiles.





## 3 Observational results


Due to the broad bandwidth of the fluorescence channel and the low intensity of the fluorescence signal, measurements were only possible during the night. At daytime, scattered solar radiation would cause too much noise in the fluorescence channel. As the MARTHA system is operated manually, the number of measurements remains limited. Since August 2022, about 50 measurements have been performed, providing more than 250 hours of atmospheric fluorescence observations. Typical atmo-


spheric values of the fluorescence backscatter coefficient and fluorescence capacity, that were obtained at Leipzig during the time period from August 2022 to October 2023, are presented in the next paragraph.

In general, $\beta_\mathrm{F}$ ranged between $1 \times 10^{-5}\,\mathrm{Mm}^{-1}\,\mathrm{sr}^{-1}$ for background aerosol and more than $1 \times 10^{-3}\,\mathrm{Mm}^{-1}\,\mathrm{sr}^{-1}$ for optically extraordinarily thick wildfire smoke layers. Correspondingly, $G_\mathrm{F}$ varied from $1 \times 10^{-5}$ for background aerosol ($\sim 10^{-6}$ for clouds) to $1.3 \times 10^{-3}$, whereas most of the measurement points were in the range of $5 \times 10^{-5}$ to $7 \times 10^{-4}$. I.e., the fluores-


cence backscatter coefficient was about four orders of magnitude lower than the elastic ones, which agrees with the findings by Veselovskii et al. (2020).

In the following, four interesting case studies are presented in several subsections. In Sec. 3.1, the fluorescence properties of wildfire smoke are discussed by analyzing an optically and geometrically thick smoke layer on 4 July 2023. In Sec. 3.2,


we first demonstrate the ability of the fluorescence lidar technique to detect optically thin aerosol layers by presenting two case studies (Sec. 3.2.1 and Sec. 3.2.2). Subsequently, we discuss the reasons for the increased sensitivity of the fluorescence channel to aerosol particles in Sec. 3.2.3. Finally, we underline the importance of this new capability by presenting a striking smoke-cirrus interaction case study in Sec. 3.2.4.

### 3.1 Fluorescence of wildfire smoke – 4 July 2023


In the spring and summer of 2023, huge wildfires raged across Canada, with unusual intensity in the provinces of Alberta and British Columbia. With the prevailing westerly winds, large amounts of biomass-burning aerosol were transported towards Europe. As a result, we frequently observed wildfire smoke layers over Leipzig from mid-May to mid-July 2023.

As a first example, the fluorescence of an optically thick plume of wildfire smoke on 4–5 July 2023 shall be characterized. Figure 2 displays the height-time distributions of the particle backscatter coefficient at 532 nm, the fluorescence backscat-


ter coefficient and the particle depolarization ratio at 532 nm for this night. Figure 2(a) shows a highly polluted troposphere (overall aerosol optical depth (AOD) of around 0.8 at 532 nm). This agrees well with data from the Aerosol Robotic Network (AERONET), where AOD values of around 0.75–0.8 were retrieved at 500 nm and 17 UTC on 4 July 2023. An optically thick aerosol layer extended from 3.4 to 5.8 km height. To determine its optical properties, a 1-hour time period was considered for temporal averaging. Figure 2(d) shows the vertical profiles of the fluorescence and elastic backscatter coefficients, together


with the fluorescence capacity averaged over the time period from 21 to 22 UTC. At the optically thickest part, $\beta_{532}$ reached values of up to $5\,\mathrm{Mm}^{-1}\,\mathrm{sr}^{-1}$. The 532 nm AOD of the whole layer amounted to around 0.48.

The optical properties of this aerosol layer were then used to determine the aerosol type. The fact that the lidar ratio at 532 nm



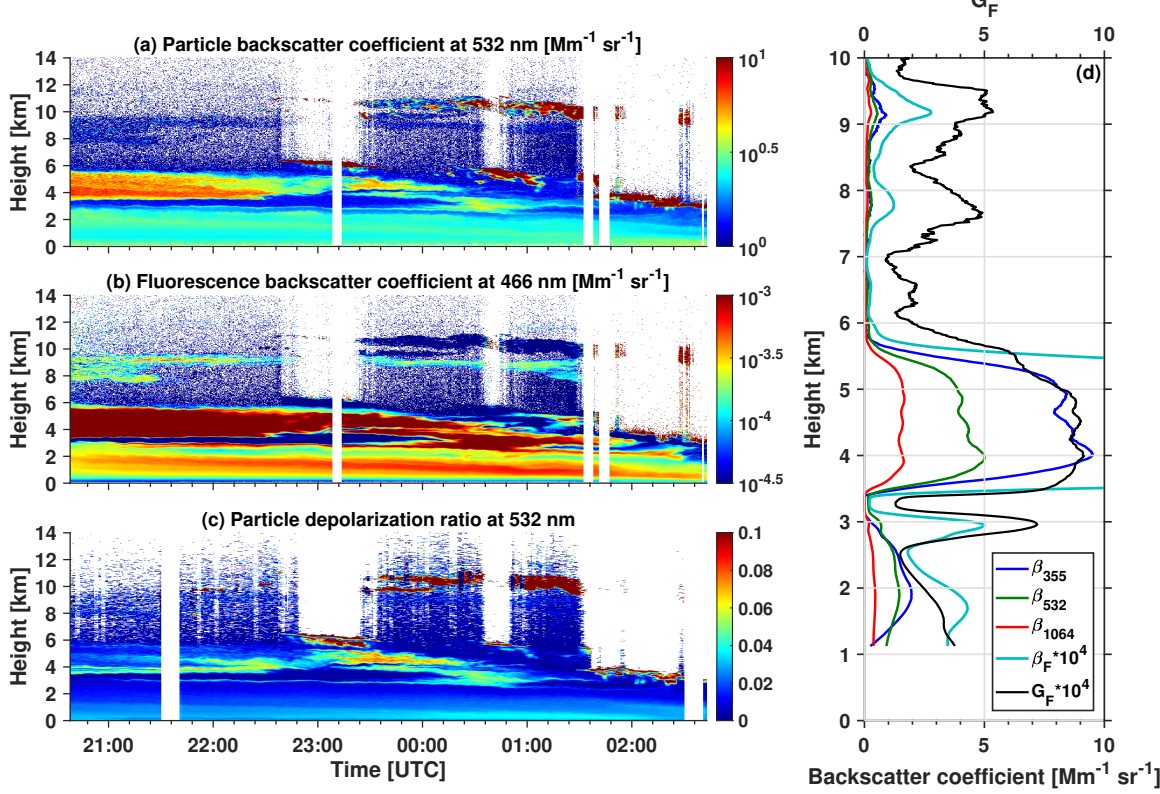

**Figure 2.** Height-time distributions of (a) particle backscatter coefficient at 532 nm and (b) fluorescence backscatter coefficient ($\beta_F$) measured with the MARTHA system and (c) particle depolarization ratio at 532 nm from Polly$^{XT}$ on 4–5 July 2023. (d) Vertical profiles of $\beta_F$ and the elastic backscatter coefficients together with the fluorescence capacity ($G_F$) from 21:00 to 22:00 UTC on 4 July 2023.

(60 sr) was significantly larger than the one at 355 nm (38 sr) and the high backscatter-related Ångström exponent (1.66) are characteristic for aged BB aerosol (Müller et al., 2005; Ansmann et al., 2009; Ohneiser et al., 2021, 2022; Hu et al., 2022;

Janicka et al., 2023). Furthermore, these retrieved lidar ratio values are in the same range as reported for aged wildfire smoke in previous studies (e.g., Murayama et al., 2004; Ansmann et al., 2009; Haarig et al., 2018; Hu et al., 2019). The low particle depolarization ratio ($\delta_{532} \leq 0.07$, cf. Fig. 2(c)) in the layer from 3.4 to 5.8 km height points to a spherical shape of the particles, which is also typical for aged wildfire smoke in the middle free troposphere (Haarig et al., 2018). Thus, it can be concluded that this aerosol layer consisted of aged BB aerosol particles.

Figs. 2(b) and (d) show a very high fluorescence backscatter coefficient ($\beta_F \approx 2.75 \times 10^{-3}\,\mathrm{Mm}^{-1}\,\mathrm{sr}^{-1}$) for this smoke layer and a corresponding fluorescence capacity of $G_F \approx 7.8 \times 10^{-4}$. In other words, smoke shows very high values of fluorescence capacity compared to other particle types and can thus be clearly identified through this new quantity. These values and the features witnessed in our observations corroborate the findings by Reichardt et al. (2018) and Veselovskii et al. (2020).



Considering the entire 2023 wildfire season, the fluorescence capacity of smoke varied from $1$–$13 \times 10^{-4}$. Thereby, values of
$2$–$7 \times 10^{-4}$ were observed most frequently, which agrees with the results of Hu et al. (2022) and Veselovskii et al. (2022a), who reported values in the range of $1$–$4.5 \times 10^{-4}$ for their observations at Lille, France. The particle depolarization ratio at 532 nm was low (below 0.07) for most (95 %) of the investigated smoke layers.

## 3.2 Detection of optically thin aerosol layers with the fluorescence channel

Besides its relevance for aerosol type identification, our results suggest an additional capability of a fluorescence lidar: to detect optically thin aerosol layers. In several measurements with the new fluorescence channel, an enhanced fluorescence signal revealed the presence of aerosol layers that went unnoticed when employing only the elastic-backscatter detection channels. Three exemplary measurement cases are discussed in the following sections.

### 3.2.1 Hidden smoke layers – 21 September 2022

Figure 3 shows the height-time distributions of the range-corrected lidar signal at 1064 nm (a), the fluorescence backscatter coefficient (b) and the fluorescence capacity (c). According to the elastic backscatter signal in Fig. 3(a), the upper troposphere appears to be rather aerosol-free. Only the polluted boundary layer, some thin layers up to 4 km height and a thin cloud at around 4 km height from 21 to 22 UTC indicated aerosol presence. However, an enhanced fluorescence backscatter coefficient in Fig. 3(b) reveals several other fluorescing aerosol structures throughout the middle and upper troposphere (at around 5, 6.5,
9 and 9.75 km height). This already illustrates already that with measurements of aerosol fluorescence, thin aerosol layers can be identified more easily from lidar quicklooks and therefore chosen for detailed analysis. Looking at the vertical profiles, this measurement case appears even more impressive. Figure 3(d) displays the time-averaged vertical profiles of the fluorescence and elastic backscatter coefficients together with the fluorescence capacity. The profiles were averaged over the 2-hour time period from 19:04 to 21:04 UTC to exclude the cloud, which was present at around 4 km height from that point onwards.
The lowest (3.3 km) and most fluorescent ($\beta_{\mathrm{F}} \approx 2.5 \times 10^{-5}\,\mathrm{Mm}^{-1}\,\mathrm{sr}^{-1}$) layer above the boundary layer still shows clearly enhanced elastic backscatter coefficients at all three wavelengths. In the mid-level layers at around 5 and 6.5 km height, the 532 nm backscatter coefficient is already only slightly enhanced compared to the background and the 355 nm backscatter coefficient even fails to resolve the aerosol layer at 6.5 km. Only the backscatter coefficient at 1064 nm shows clear maxima for these aerosol layers. However, the elastic-backscatter detection channels reach their limits with the two high layers at 9 and
9.75 km altitude. While the 355 nm backscatter coefficient is completely noisy in this altitude range, $\beta_{532}$ and $\beta_{1064}$ do show maxima in the altitude range of the increased $\beta_{\mathrm{F}}$. But these maxima are difficult to distinguish from the background, which is likewise already quite noisy. Thus, it is unlikely that these two higher layers would have been detected as aerosol layers without the additional fluorescence information (also because the particle depolarization ratio is also quite low with 2 %, not shown). The overall AOD of this measurement case was around 0.13 at 532 nm (again in agreement with AERONET: 0.1 at 500 nm),
whereas the majority of the aerosol was found in the boundary layer (AOD $\approx$ 0.1). The smoke layers above the boundary layer




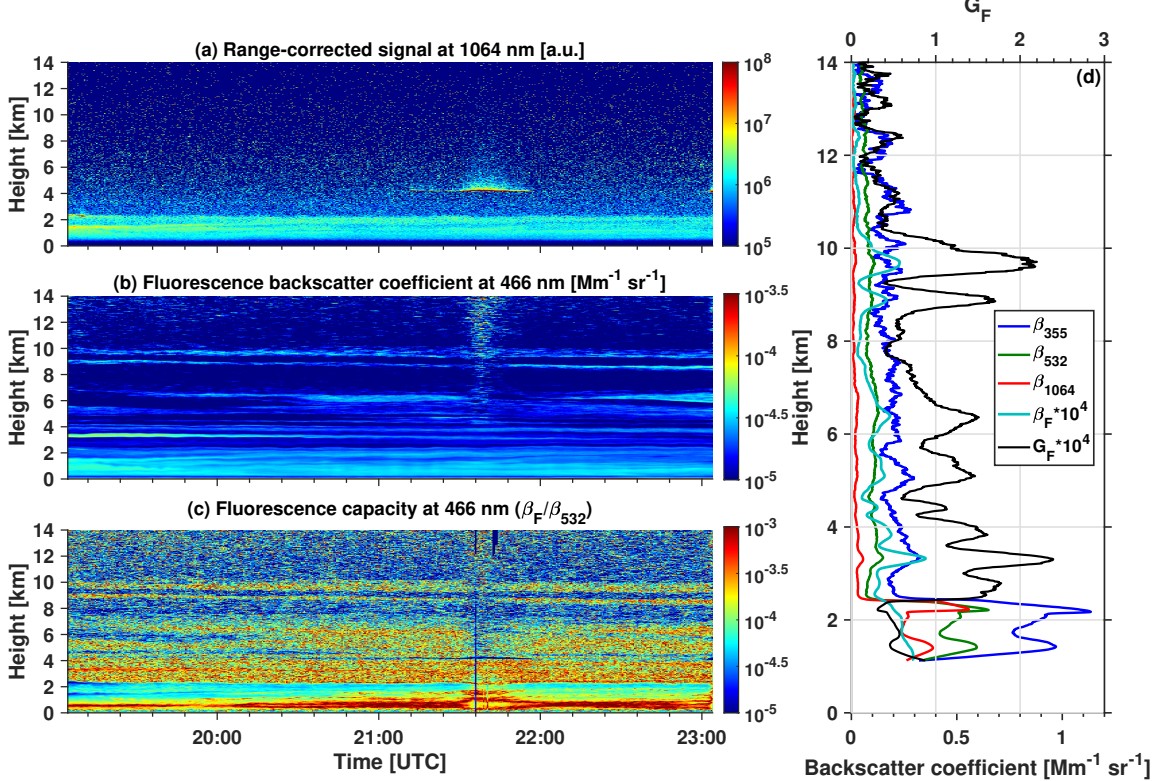

**Figure 3.** Height-time distributions of (a) range-corrected lidar signal at 1064 nm, (b) fluorescence backscatter coefficient ($\beta_F$) and (c) fluorescence capacity measured with the MARTHA system on 21 September 2022. (d) Vertical profiles of $\beta_F$ and the elastic backscatter coefficients together with the fluorescence capacity ($G_F$) from 19:04 to 21:04 UTC on 21 September 2022.

only added up to an AOD of around 0.03. The two thinnest layers at around 9 and 9.75 km height even had an AOD of only 0.002 each at 532 nm.

### 3.2.2 A thin smoke layer in the UTLS – 15 May 2023

Another example of such "unnoticeable" layers is the night from 15-16 May 2023. Figure 4(a-c) displays the height-time dis-
tributions of the range-corrected lidar signal at 532 nm, the fluorescence backscatter coefficient and the particle depolarization ratio at 532 nm. The vertical profiles of the backscatter coefficients together with the fluorescence capacity for the period from 01:15 – 02:15 UTC are shown in Fig. 4(d). This measurement case is characterized by a pronounced fluorescent aerosol layer (532 nm AOD $\approx$ 0.05), ranging from 4 to 6.7 km height. The high fluorescence capacity ($G_F \approx 5 \times 10^{-4}$) allows us to identify the aerosol particles present as wildfire smoke. The particle depolarization ratio at 532 nm is low (around 1.7 %), indicating a
well-advanced aging process of the smoke particles. At around 11 km height, the range-corrected signal at 532 nm in Fig. 4(a)





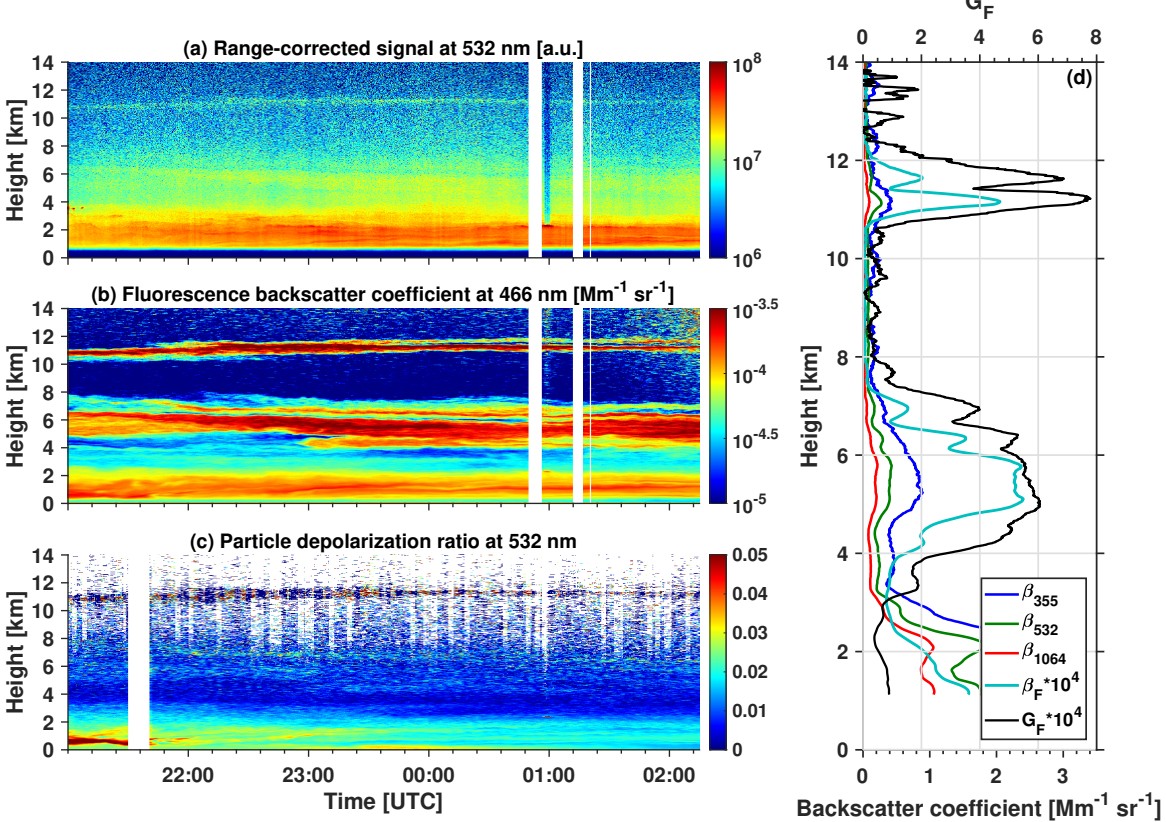

**Figure 4.** Height-time distributions of (a) range-corrected lidar signal at 532 nm and (b) fluorescence backscatter coefficient ($\beta_\mathrm{F}$) measured with the MARTHA system and (c) particle depolarization ratio at 532 nm from Polly$^\mathrm{XT}$ on 15–16 May 2023. (d) Vertical profiles of $\beta_\mathrm{F}$ and the elastic backscatter coefficients together with the fluorescence capacity ($G_\mathrm{F}$) from 01:15 to 02:15 UTC on 16 May 2023.

shows another highly fluorescent smoke layer ($G_\mathrm{F} \approx 6.5 \times 10^{-4}$). This higher value of the fluorescence capacity indicates a more efficient fluorescence emission in this aerosol layer than in the lower one. The reason for this remains unclear. On the one hand, this could be purer smoke, while the lower layer could also contain a small proportion of another less fluorescent aerosol type. On the other hand, the BB aerosol in both layers could differ in chemical composition due to different fire sources.
However, backward trajectory analyses rather oppose these ideas, showing similar pathways and pointing to the same source region (the northern part of the North American continent) for both altitudes. Thus, the aforementioned Canadian wildfires can be assumed as the corresponding smoke sources. Hence, the difference in the fluorescence capacities can be attributed to the distinct environmental conditions during atmospheric transport. At the upper smoke layer, the particle depolarization ratio was slightly enhanced ($\delta_{532} \approx 6.5\,\%$) compared to the lower layer. This indicates a more irregular shape of the particles (although
in general terms this is still almost spherical), meaning that the aging process had not yet progressed as far as in the layer at around 6 km height, where the aerosol particles may have swollen to spherical shapes due to the adsorption of water vapor





from the ambient air (Haarig et al., 2018; Ansmann et al., 2021). With this hygroscopic growth, the elastic backscattering may have enhanced, which may have decreased the fluorescence capacity (Veselovskii et al., 2023). If a liquid shell has developed around the smoke particles, it may even have reduced the fluorescence emission of the smoke particles, an effect also known as

fluorescence quenching (e.g., Lakowicz, 2006). Due to the lower humidity and temperature at higher altitudes, the aging may have proceeded more slowly in the upper aerosol layer, so that the smoke particles maintained a more irregular shape (Knopf et al., 2018; Ansmann et al., 2021) and a higher fluorescence capacity.

Furthermore, Fig. 4(b) reveals another aerosol layer with enhanced fluorescence directly above at around 11.7 km. The complete structure of this layer remains unnoticed in the range-corrected signal in Fig. 4(a). This impression is confirmed by the

vertical profiles in Fig. 4(d). The thin aerosol layer at around 11.7 km cannot be distinguished from the background noise in $\beta_{355}$. $\beta_{532}$ and $\beta_{1064}$ exhibit a slight increase, although this increase is only very weakly pronounced at 1064 nm. So, only the 532 nm backscatter coefficient shows a clear peak for this layer. At a closer look, also the time-height distribution of the particle depolarization ratio at 532 nm in Fig. 4(c) indicates this layer by slightly increased values at this altitude. But again, it would have been hard to recognize this layer from the elastic backscattering products alone, without having a clearer pic-

ture of the aerosol situation from the fluorescence channel. This underlines the potential of the fluorescence lidar technique beyond aerosol characterization. Fluorescence backscatter can be used for the detection of aerosol layers in scenarios where concentrations are below the lower detection limit of the elastic-backscatter channels.

### 3.2.3   On the capabilities of a dedicated fluorescence channel

The example cases presented above have clearly shown the advantages of adding fluorescence observations to the analysis. In

the following, these enhanced capabilities are discussed.

The three elastic-backscatter channels rely on the principle of elastic backscattering of the emitted laser radiation, which occurs at both air molecules and aerosol particles. As a result, the received backscattered signal intensity is the sum of the radiation scattered by air molecules and aerosol particles. Figure 5(a) illustrates this context by showing the molecular (blue curve), particle (magenta curve) and total (molecular + particle, green curve) backscatter coefficient at 532 nm for the time

period from 01:15 to 02:15 UTC on 16 May 2023. Due to the strong reduction of the air density with height, the molecular scattering intensity steadily decreases with height. Aerosol particles are usually most concentrated close to the ground and tend to decrease with height as well, although this is not necessarily always the case. The resulting wide range of values, together with the inherent signal reduction due to the solid angle of the telescope ($\propto \frac{1}{R^2}$) and the atmospheric transmission (exponential term), leads to a large dynamical range of the received elastic backscatter lidar signal (i.e., a strong reduction with height). This

is illustrated by the plot of the range-corrected elastic-backscatter lidar signal at 532 nm (green curve in Fig. 5(b)). It shows strong backscattering at lower altitudes and much weaker signals at higher altitudes. To avoid the saturation of the detectors at low altitudes, usually ND filters are used to reduce the signal. This signal reduction mainly decreases the sensitivity at high altitude ranges, hampering in this way the detection of thin aerosol layers in the UTLS region.

The reason for the low return signals in the first kilometer in Fig. 5(b) is the overlap effect. Due to the geometry of the receiving

telescope, only a part of the actual (strong) backscatter signal can be imaged onto the detector at the lowest altitudes. A so-




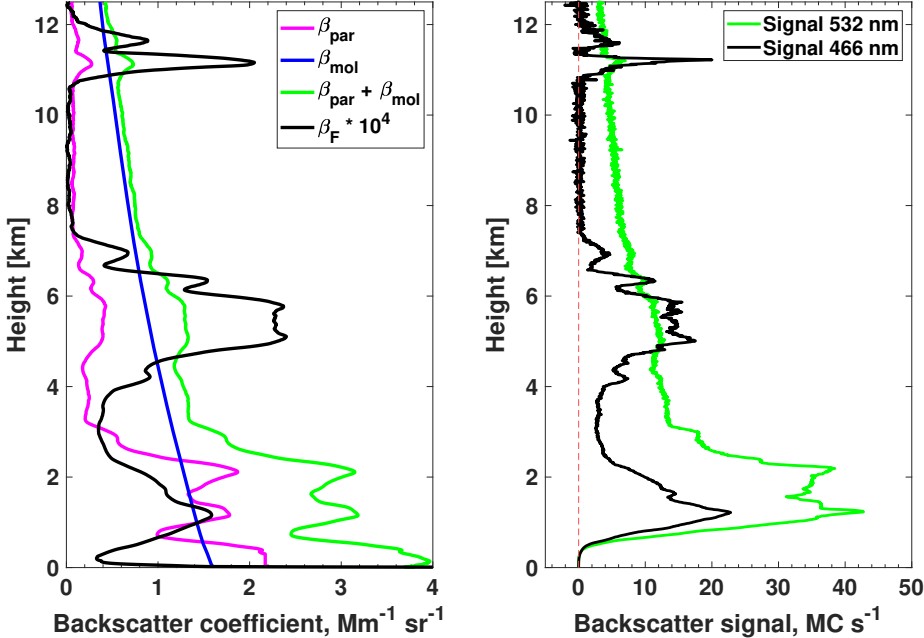

**Figure 5.** Vertical profiles of (a) particle (magenta), molecular (blue) and total (green) backscatter coefficients at 532 nm and fluorescence backscatter coefficient (black) and (b) range-corrected signals at 532 nm (green) and 466 nm (black) on 16 May 2023 for the time period from 01:15 to 02:15 UTC.

called overlap function was previously determined and used to correct for the overlap effect in the retrieval of the backscatter coefficients so that they can be determined almost down to the ground (cf. Fig. 5(a)). The total elastic backscatter coefficient (green curve in Fig. 5(a)) clearly shows that the strongest elastic backscattering is observed at low altitudes and near the ground. Instead, the fluorescence backscatter coefficient (black curve in Fig. 5(a)) is not necessarily peaking at the lowest altitudes. In

this case, it is strongest at around 4.5 to 6 km height. This is because fluorescence backscattering is only produced by aerosol particles, as air molecules do not allow fluorescence transitions. As a result, the dynamic range that the detector needs to cover is considerably reduced. This can be visualized in Fig. 5(b) in terms of real measurements. In aerosol-free altitude ranges (e.g., from 8 to 10 km in this case), the range-corrected fluorescence signal (black curve) is nearly 0 and only increases in the presence of fluorescing aerosol particles. Therefore, the dynamical range of the fluorescence channel is much smaller, and

because of that, the fluorescence channel can usually be operated without any ND filter, which further increases its sensitivity to thin aerosol layers at higher altitudes, as can be seen for the aerosol layers at 11 to 12 km height.





### 3.2.4 Atmospheric implication: Smoke-cirrus interaction – 29 May 2023

Now, after discussing the possibility of detecting such thin aerosol layers, the question of their relevance in atmospheric research arises. Because of their low optical thicknesses (of typically $\leq 0.01$), such aerosol layers might not have a relevant
radiative effect, but they may impact cloud formation, e.g., by serving as INPs. In both cases presented above (Sec. 3.2.1 and 3.2.2), the measurements of the fluorescence backscatter coefficient revealed thin wildfire smoke layers at rather high altitudes around the tropopause. This altitude range, also referred to as upper troposphere and lower stratosphere (UTLS) region, is a common site for the formation of cirrus clouds. And indeed, several measurement cases during the 2023 wildfire season showed cirrus clouds directly below such thin smoke layers. One example (29-30 May 2023) is displayed in Fig. 6 and will be
discussed in the following.

The range-corrected lidar signal at 532 nm in Fig. 6(a) shows cirrus clouds, that extended from 7 to 11.5 km at the beginning of the measurement. Above, enhanced values of the fluorescence backscatter coefficient (see Fig. 6(b)) reveal the presence of a smoke layer at 10.5 to 12 km height, that was not visible in the elastic-backscatter lidar signal over large parts of the observation period. Only at the end of this measurement (around 02:00 UTC), when the clouds became thinner and more scattered, the
aerosol layer could be anticipated vaguely from weak signatures in the range-corrected signal (cf. Fig. 6(a)).

Remarkably, the upper boundary of the cirrus clouds coincides with the lower boundary of the fluorescing smoke layer for large parts of the observation period. The elastic-backscatter signal in Fig. 6(a) clearly shows pronounced virga structures (i.e., stripes of falling ice crystals). Such an arrangement has already been reported in the literature for smoke layers and cirrus clouds observed over the eastern Mediterranean and in the Arctic (Mamouri et al., 2023; Ansmann et al., 2024b).

This case reveals another feature of fluorescence backscattering that is exclusive to aerosol particles. Water content can quench the fluorescence scattering, and, as our measurements from 2022–2023 showed, hydrometeors such as droplets and ice crystals exhibit the lowest values of fluorescence capacity. This feature has also been pointed out in previous studies (Reichardt et al., 2018; Veselovskii et al., 2022a) and opens a new door into aerosol and cloud detection. In combination, elastic-backscatter and fluorescence channels can unambiguously separate aerosol particles and hydrometeors that are coexisting within the same air
volume.

The arrangement of the cloud and the aerosol layer in this case indicates that the ice nucleation happened at the cloud top, from where the freshly formed ice crystals were falling down, thus producing the aforementioned falling stripes. Furthermore, the smoke layer slowly rose in altitude towards the end of the measurement. At the same time, the cloud top rose first, and later, the clouds even became scattered and seemed to dissolve. All these facts indicate that the smoke particles may have triggered the
cloud formation by serving as INPs. However, the ability and relevance of smoke particles to act as INPs is an open question in the literature. Although a few available observations showed enhanced immersion-mode INP concentrations inside of BB aerosol plumes (Barry et al., 2021; McCluskey et al., 2014), wildfire smoke is considered to be a rather inefficient INP at temperatures above -30 °C compared to other aerosol types such as dust (e.g., Barry et al., 2021; Knopf et al., 2018). Thus, BB aerosol is, in general, not considered a relevant INP source in mixed-phase cloud processes. Likewise, in situ assessments
have suggested that BB aerosol particles rarely freeze to form cirrus clouds (Froyd et al., 2009, 2010). However, the authors





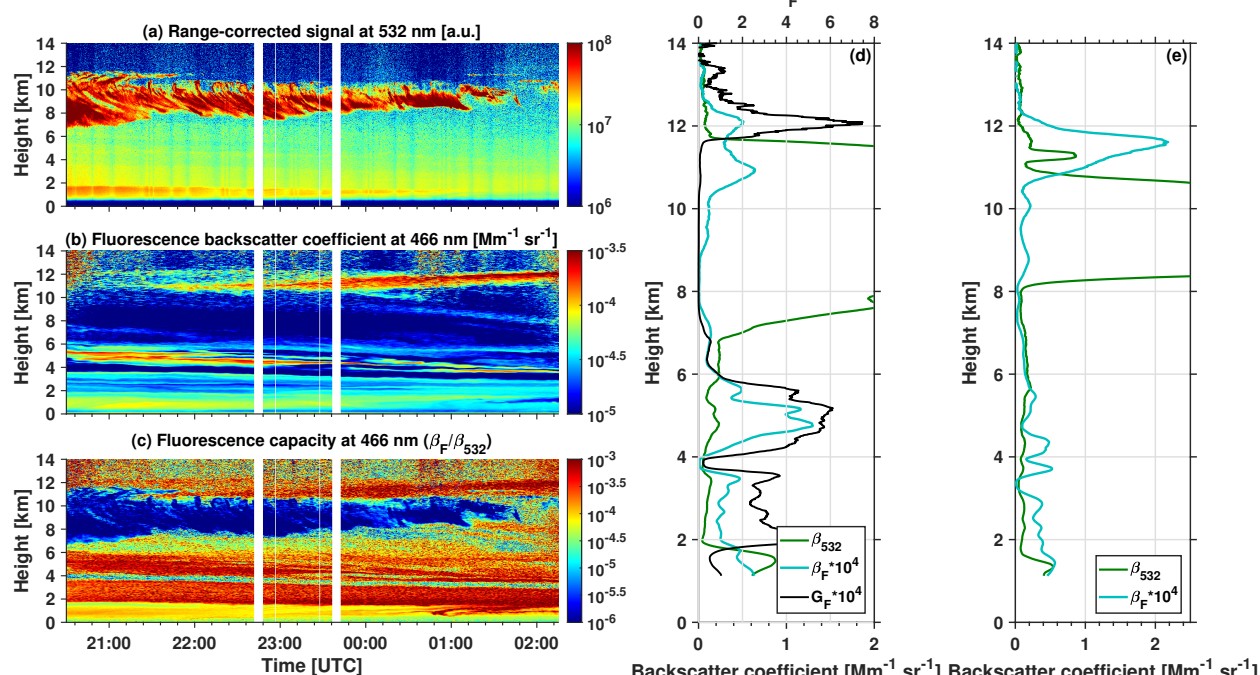

**Figure 6.** Height-time distributions of (a) range-corrected lidar signal at 532 nm, (b) fluorescence backscatter coefficient ($\beta_F$) and (c) fluorescence capacity measured with the MARTHA system on 29 - 30 May 2023. Vertical profiles of $\beta_F$ and the elastic backscatter coefficient at 532 nm together with the fluorescence capacity ($G_F$) from (d) 21:00 to 22:00 UTC on 29 May 2023 and (e) 0:15 to 1:15 UTC on 30 May 2023.

could not exclude the INP ability of BB particles due to temperature limitations in their experimental setup. Recent lidar-based studies discussed the potential of smoke particles to promote freezing via deposition and provided evidence of BB aerosol acting as the main INP source in cirrus clouds observed at Limassol, Cyprus (Mamouri et al., 2023) and in the Arctic (Ansmann et al., 2024a, b). Simulations considering gravity waves further explain how heterogeneous freezing overtakes the main role,
consuming quickly the water vapor and reducing supersaturation, hampering in this way homogeneous freezing (Ansmann et al., 2024a). Investigations of possible smoke-cirrus interactions in the UTLS region will be an important topic for future studies.

At the beginning of the measurement on 29 May, parts of the cirrus clouds were even embedded in the smoke layer. Fig-
ure 6(d) displays the vertical profiles of the elastic and fluorescence backscatter coefficients, together with the fluorescence capacity from 21 to 22 UTC. From the high elastic backscatter coefficients, two nucleation sections can be identified. There is a lower part with ice crystals falling from about 10.25 km, and an upper part where ice crystals start falling from 11.75 km. These falling ranges somehow coincide with the aerosol layers observed with the fluorescence channel. Cloud-top temperatures





were estimated from radiosondes launched at the nearest station Lindenberg (150 km away) and ranged from -55 to -51 °C, a
temperature range in which deposition ice nucleation is particularly efficient (Ansmann et al., 2024a). Above, the high fluo-
rescence capacity (up to $G_\mathrm{F} = 7.5 \times 10^{-4}$ at the maximum) indicates the smoke layer. However, the fluorescence backscatter
coefficient shows enhanced values over a wider altitude range, even down to the upper boundary of the lower cloud part, which
supports the hypothesis that wildfire smoke particles triggered the ice cloud formation. An interesting feature in Fig. 6(d) is
the reduction of the fluorescence backscatter at the cloud top of the upper cirrus part ($\beta_\mathrm{F} \approx 2.9 \times 10^{-5}\,\mathrm{Mm}^{-1}\,\mathrm{sr}^{-1}$) compared
to the higher values above this upper cloud layer ($\beta_\mathrm{F} \approx 5 \times 10^{-5}\,\mathrm{Mm}^{-1}\,\mathrm{sr}^{-1}$) and at the top of the lower part of the cirrus
cloud ($\beta_\mathrm{F} \approx 6.4 \times 10^{-5}\,\mathrm{Mm}^{-1}\,\mathrm{sr}^{-1}$). A possible reason for this reduction could be fluorescence quenching by the ice crystals
inside the upper cloud layer. There is static quenching (the formation of a non-fluorescing complex from the fluorophore and
a quencher), dynamic quenching (the collision of the fluorophore with a quenching molecule) and resonance energy transfer
(Lakowicz, 2006). Water is known to act as a fluorescence quencher for organic fluorophores (e.g., Stryer, 1966; Dobretsov
et al., 2014). However, it has to be stated that these studies considered fluorophores in aqueous solutions and not enclosed in ice
crystals. Besides, studies reported that fluorescence quenching by water is strongest for red-emitting fluorophores, where it can
reduce the fluorescence emission by up to a factor of 3 (Maillard et al., 2021). For shorter absorption and emission wavelengths
the effect is much weaker (Fürstenberg, 2017; Maillard et al., 2021).

Another reduction of the fluorescence backscatter coefficient was observed inside the ice virga. In the middle part of the cloud,
between 8 and 10 km, $\beta_\mathrm{F}$ ranged at minimal values of about $1 \times 10^{-6}$ to $1 \times 10^{-5}\,\mathrm{Mm}^{-1}\,\mathrm{sr}^{-1}$. Aerosol scavenging arises as
a possible explanation. I.e., after ice nucleation, the freshly formed ice crystals fell down and collected most of the aerosol
particles (impaction), reducing the aerosol load in the cloud layer. In this case, one would expect an accumulation of smoke
particles at or directly below the cloud base. Indeed, near the cloud base at around 7 km height, the fluorescence backscatter
increases again up to $1.4 \times 10^{-5}\,\mathrm{Mm}^{-1}\,\mathrm{sr}^{-1}$. However, both reductions discussed here could also be due to different aerosol
loads and characteristics at the different altitudes. The situation is, in any case, complex, and further investigations of similar
cases are needed to characterize aerosol particles inside clouds by fluorescence observations.

In summary, our measurement results suggest two possible interaction pathways between the observed smoke layer and the
cirrus clouds: fluorescence quenching and heterogeneous ice nucleation. For further illustration, Fig. 7 shows the elastic and
fluorescence backscatter coefficients together in one plot. The height-time bins with pronounced aerosol fluorescence (in gray
color) along with the elastic backscattering at 532 nm clearly show a major aerosol-cloud interplay. Just before 1 UTC, an
interesting situation arose. The smoke particles were deeply embedded in the cloud, exhibiting two layers: one around 10 km
and one between 8 and 9 km, accompanied by a significant increase in the elastic backscatter coefficient.

A further potential application of fluorescence lidar is to provide INP information in such cases with low but relevant aerosol
presence in the cloud surroundings, especially at the cloud top. A conversion from the unambiguous fluorescence backscatter
coefficient to an INP number concentration ($N_\mathrm{INP}$) is desirable. An approach applying conversion factors, which link the fluo-
rescence backscatter coefficient with the previously inverted microphysical properties of the fluorescing aerosol particles from





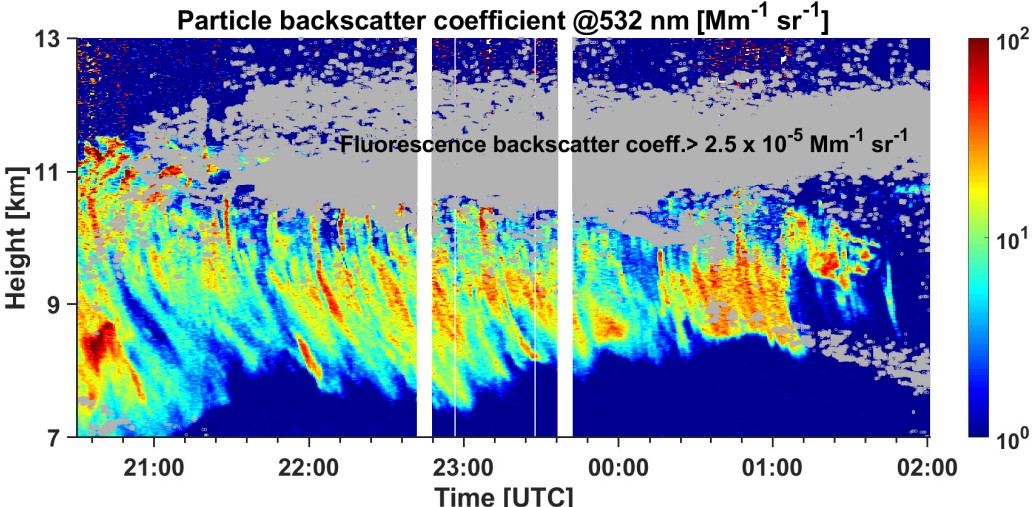

**Figure 7.** Height-time distributions of the particle backscatter coefficient at 532 nm during the night from 29 - 30 May 2023. Height-time bins with a large fluorescence backscatter coefficient ($> 2.5 \times 10^{-5}$ Mm$^{-1}$ sr$^{-1}$) are colored gray.

multi-wavelength lidar data, was suggested by Veselovskii et al. (2022a). In the case of a low aerosol load or inside a cloud layer, the resulting mean conversion factors, together with the fluorescence backscatter coefficient, can then be used to derive the aerosol surface area concentration, which is needed as input to the INP parameterization (Veselovskii et al., 2022a). An alternative approach would be to determine $N_{\mathrm{INP}}$ directly from ice crystal number information provided by lidar-radar synergy and find a conversion factor between $\beta_{\mathrm{F}}$ and $N_{\mathrm{INP}}$. Such a factor $\sigma_{\mathrm{F}}$ would be in the form of $\sigma_{\mathrm{F}} = N_{\mathrm{INP}}/\beta_{\mathrm{F}}$ and could be used for cirrus cloud scenes with comparable temperature and humidity. Preliminary assessments of INP concentrations via the POLIPHON method (Ansmann et al., 2012; Mamouri and Ansmann, 2014) and ice crystal number concentrations from lidar-radar synergy (Bühl et al., 2019) suggest a conversion factor in the range of 3–8 $\times 10^4\,\mathrm{Mm\,sr\,L}^{-1}$.

A reliable conversion to link the fluorescence backscatter coefficient to ice nuclei concentrations would be beneficial to investigate aerosol-cloud interactions, especially in those situations with low aerosol amounts. Further aerosol-cloud cases will be investigated in the future to evaluate this potential application of fluorescence backscatter information specifically.

## 4  Conclusions

In this study, we presented the newly-implemented fluorescence channel in the lidar system MARTHA, located at TROPOS, Leipzig, Germany. Some of the first measurements performed with the upgraded system were during the 2023 summer wildfire season. The fluorescence capacity of wildfire smoke mainly ranged between $2 \times 10^{-4}$ and $7 \times 10^{-4}$, thus confirming previously reported values in the literature (Hu et al., 2022; Veselovskii et al., 2022a).

Special care was put into the characterization of the fluorescence lidar, where each component along the optical path was considered to determine the system efficiency constants needed to derive the new fluorescence parameters. The detection of



optically thin aerosol layers that are only recognizable in the fluorescence signal can significantly improve the detection capabilities of a lidar, which could be critical for low-particle-concentration situations. The enhanced sensitivity results from the fact that laser-induced fluorescence emission originates exclusively from aerosol particles, while air molecules and hydrometeors are excluded from this scattering process. Furthermore, as our observations showed, the new dedicated "particle" channel enables an unambiguous differentiation between coexisting unactivated aerosol particles and hydrometeors within clouds.


Because of their strong fluorescence and rather low depolarization, the aerosol layers presented in the case studies could be identified as biomass-burning aerosol. The analysis of backward trajectory calculations suggested that the discussed layers consisted of smoke particles from Canadian wildfires. The measurements show that such optically thin smoke layers are not so rare in the UTLS region. This suggests that the atmosphere over Europe might be more polluted than previously thought, especially during the summer wildfire season. Those thin layers might not have a strong direct radiative impact, but at these altitudes, smoke particles could become an additional INP source in an otherwise relatively clean atmosphere. Investigating such aerosol layers with a fluorescence lidar, combined with advanced remote-sensing techniques to assess cloud microphysics, could provide more clarity about the relevance of heterogeneous freezing of smoke particles in cirrus cloud formation compared to homogeneous nucleation onto small droplets from background particles. Several observations of cirrus clouds directly below thin biomass-burning aerosol layers suggest that these might be the primary INP source, indicating that heterogeneous freezing is the dominant process. To thoroughly explore this potential aerosol-cloud effect, a larger data set would be beneficial and might provide stronger evidence and more detailed insights into this hypothesis.




Further instrumental upgrades are currently ongoing in the MARTHA system. A new powerful laser, together with a 32-channel spectrometer, will extend the observational sharpness and aim to provide state-of-the-art information about aerosol and clouds from the ground up to the stratosphere.

*Data availability.* Lidar data and products are available upon request at info@tropos.de or polly@tropos.de. The backward trajectory analysis is based on air mass transport computation with the NOAA (National Oceanic and Atmospheric Administration) HYSPLIT (HYbrid Single-Particle Lagrangian Integrated Trajectory) model (http://ready.arl.noaa.gov/HYSPLIT_traj.php). AERONET photometer observations of Leipzig are available in the AERONET database (http://aeronet.gsfc.nasa.gov/. GDAS1 (Global Data Assimilation System 1) re-analysis products from the National Weather Service's National Centers for Environmental Prediction are available at https://www.ready.noaa.gov/gdas1.php.


*Author contributions.* BG and CJ conceptualized and organized the study. BG wrote the manuscript with the help of CJ. BG took care of the fluorescence products supported by CJ, AA and HB. BG, CJ, RE, AA, UW and MH worked on the experimental setup of the new detection channel. MH, FF, AAF, HG, JH, KO, CJ and BG performed the lidar measurements. MR contributed to the monitoring of the daily aerosol-



cloud situation. PS, RE, and KO took care of the Polly$^{XT}$ lidar system. All co-authors contributed to the several discussions about the new technique and proofreading.

*Competing interests.*  The contact author has declared that none of the authors has any competing interests.

*Acknowledgements.*  For the fruitful discussions and openness about the new fluorescence technique, we would like to thank Igor Veselovskii, Qiaoyun Hu and Jens Reichardt. We acknowledge the technical team from TROPOS for the experimental support. This research has been supported by the German Federal Ministry of Education and Research (BMBF) under the FONA Strategy "Research for Sustainability" (grant no. 01LK2001A). The contribution of BG was supported from tax revenues on the basis of the budget adopted by the Saxon State
Parliament.



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
