# Peer review of "Invisible aerosol layers: improved lidar detection capabilities by means of laser-induced aerosol fluorescence"

_EGUsphere, 2024_

## Referee Comment (RC1)

**Review to "Invisible aerosol layers: improved lidar detection capabilities by means of laser-induced aerosol fluorescence" by Benedikt Gast et al.**

**General comment**

This paper demonstrates the detection capability of fluorescence lidar in revealing and identifying high-altitude smoke layers, based on the measurements collected during the Canadian wildfire season in 2023. These layers, due to their low particle concentrations and high altitudes, may be beyond the detection limits of conventional Mie-Raman lidar systems, yet they still influence the ice cloud formation and cloud properties. The authors first introduce the calibration method of the fluorescence channel, followed by an analysis of several scenarios of measurements. These results demonstrate that the fluorescence channel enhances the detection capability of a conventional Mie-Raman lidar, brings new information about aerosol characterization and provides new opportunities for the study of aerosol-cloud interactions. The study aligns well with the scope of <Atmospheric Chemistry and Physics> and the measurements presented are valuable. The presentation of data is clear and the conclusions are scientifically sound. Therefore, I recommend the paper for acceptance. However, major revisions regarding the following aspects are needed:

1. In the calibration procedure, the calculation of detector efficiency contains two terms: the PMT quantum efficiency and the PMT gain ratio. According the text in line 148-153, the quantum efficiency is determined using the data provide by Hamamatsu and the PMT gain ratio is derived by swapping the detectors in the nitrogen Raman and fluorescence channels. However, the description about how this experiment was conducted remains unclear. Please clarify and verify whether the PMT gain ratio calculation has already accounted for the ratio of quantum efficiency.

2. The writing needs improvements, as some sentence and expressions lack of the precision expected in scientific papers. The paper contains many colloquial phrases that should be avoided to ensure the writing is concise and accurate. Please refer to specific comment for details.

3. Section 3.2.4 is lengthy and lacks of clear structure, making it difficult to follow. Please consider condensing the section for better readability. The authors could begin by introducing the observations, followed by the analysis that progressively address key aspects.

**Specific comment**

- L20: A logic issue with 'By serving as cloud condensation nuclei (CCN) or ice nucleating particles (INPs)…, the microphysical properties of water clouds…'
- L41: "Laser-induced fluorescence is a known process and several remote-sensing applications are based on it." → Laser-induced fluorescence is a well-established technique, serving as the basis of several remote sensing applications.
- L43: In the atmosphere → In the domain of atmospheric research

- L51-52: "but most were small…" do you mean these lidars were with simple configuration? And please add reference.
- L111: "and only a little loss of fluorescence return" → "with minimal loss of the fluorescence return"
- L114: only if no rain is expected → only when no rain is expected
- L114: Complete nights were collected since 2022 … → Complete night measurements have been collected since 2022…
- L116: the derivation of  new products
- L117: similarly as → similarly to
- L119,L120: The transmission at elastic wavelenght is missing. It is important for the validity of lidar equations, although the 2 terms cancel out in Equation 3.
- L122: "and the $C_R$ and $C_F$ represent the corresponding lidar calibration constants."
- L210: …polluted troposphere, with overall aerosol optical depth (AOD) of around 0.8 at 532 nm.
- L212: …and 17 UTC on 04 July… → …at 17:00 UTC on 04 July 2023. Please use standard time notation and keep it consistent everywhere in the manuscript.
- Figure 2: what are the white points in lidar quicklooks?
- Figure 2(d) is too crowded and the profile of fluorescence backscatter coefficient is truncated, please split this figure into 2, for clarity and better presentation.
- L217-220: Please indicate the vertical range which you are describing. Such lidar ratios were detected in tropospheric layer or UTLS layer?
- L243: 22 UTC…. indicated aerosol presence→ 22:00 UTC…were presented in the atmosphere. Just for logic consistency, since a cloud layer also existed.
- L245: This already illustrates  that with measurements…
- L252: the 532 nm backscatter coefficient is  only slightly enhanced
- Figure 3: the curves are too crowded, if you split them into 2: elastic backscattering + fluorescence backscatter with capacity, it will look better. And the peak at about 2.2 km seems to be cloud at around 19:10 UTC.
- L251-257: It seems, in this case, the detection ability of elastic channels is already at their limit. The description about maxima is not quite evident according to the figure. How did you smooth the profiles? Is the smoothing method wavelength dependent or vertically varying? How come the backscattering at 532 fails to resolve the aerosol layer below 6.5 km but is still able to resolve the maxima at above 9 km?
This case shows the detection limit of 532 nm backcattering is about 0.2 $Mm^{-1}sr^{-1}$, is it fair to say so? Do you have any comment about the vertical variation of the fluorescence capacity? Are they real and link to smoke properties/compositions or more likely to be artifacts produced by the vertical variation of elastic and/or fluorescencence backscattering and/or smoothing method?

- L258: (also because the particle depolarization ratio is also quite low with 2 %, not shown) → especially because the particle depolarization ratio (not shown) is also quite low, around 2 %.
- L269: (again in agreement with AERONET: 0.1 at 500 nm)→ which is consistent with AERONET data showing 0.1 at 500 nm
- L264: Another example of such "unnoticeable" layers is the night of () 15-16 May 2023. Similar error in L266.
- L268: In the range of 4-6.7 km, the vertical variation of fluorescence capacity is significant, from $2*10^{-4}$ to $6*10^{-4}$, such strong variation is also observed in the layer at 10.5-12.2 km. The comparability of fluorescence capacity between upper layer and

lower layer depends on the selection of vertical range. So, again, the question is: do you think this variation of fluorescence capacity is real? Why would it change so much in the same plume? Is there any explanation for the drop of fluorescence capacity at the edges of the plume? Or it is just some artifacts arising from different sensitivity of detection channel or smoothing ?

- L269,270: What is the travel time of this smoke layer, at 4-6.7 km? And how many days does it need to become so spherical? What is the relative humidity in this layer? The elastic backscatter coefficeint is so low in this layer that there appears to be no evidence of hygroscopic growth. Does lidar ratio tell something?

- L275: Does the back trajectory reveal any difference in transport time or pathway between the two layers?

- L276: 'the aforemential Canadian wildfires' : not clear which wildfire is it referring to

- L288: '…reveals another aerosol layer": this layer is also called "another ..layer" in L271. Better to name the layers to be 'Layer 1, 2,…' in each case to improve the clarity.

- L307-309: This long sentence should better be rephrased to improve clarity.

- L310: Is it range-corrected signal in Figure 5(b)? Is the fluorescence signal rescaled? What is the definition of the unit 'MC*s$^{-1}$'? Are they from analog channels or photon-counting channels?

- L315: only a part of the actual (strong) backscatter signal: (strong) ? this is confusing.

- L316,317: please provide the reference of the overlap correction. Are the profiles shown in previous figures corrected from overlap? Why are they cut below 1 km?

- L324: dynamical → dynamic

- L324-325: …much smaller, and because of that, the…→ …much smaller. Consequently, the…

- L332: 'UTLS' defined twice

- Figure 6: please add the profiles of temperature and relative humidity with respect to ice.

- L346: "water content can quench fluorescence scattering": this sentence is not clear enough. How does water content influence fluorescence capacity, by supressing fluorescence emission or enhancing elastic scattering?

- L349: separate → differentiate

- L352,353: 'Furthermore, the smoke layer rose…the cloud top rose first, and later, become scattered and seemed to dissolve. All these facts indicate that the smoke particles may have triggered the cloud formation by serving as INPs'. The link between smoke particles serving as INPs and the movements of clouds and smoke layers is not clearly explained here. In addition, as clouds became thinner and thinner, the smoke layers seemed to get thicker, is it related to ice crystal formation?

- L354: "the clouds even became scattered and seemed to dissolve": why do ice clouds dissolve at such temperature?

- L369: on 29 May→ in the night of 29 May

- L371: 21 to 22 UTC → 21:00 to 22:00 UTC

- L371: please indicate the two nucleation sections in Figure 6(d)

- L375: 'Above, the high fluorescence capacity…': In the context, 'Above' does not specify what you are referring to.

- L376-382: A general comment regarding the description of figures: when presenting values in a vertical profile, please include their corresponding vertical levels rather than relying only on descriptive phrases like 'down to the upper boundary of the lower cloud part' or 'at the top of the lower part of the cirrus clouds'…

- L382-383: If this difference in fluorescence backscattering can be explained by fluorescence quenching, then why did it occur specifically at the cloud top of the upper cirrus part, rather than elsewhere?

- L390: 'ranged from (at) the minimal values of…to…'

- L392--394:  ' In this case,…again up to $1.4*10^{-5}Mm^{-1}sr^{-1}$ .'' --- the fluroescent layer at around 7 km seemed not in contact with cirrus clouds after 21:00, and there was no fluorescence between 7.5 km and 8.5 km. Therefore, I am not convinced that this layer was due to the accumulation of smoke particles at cloud base.

- Figure 6(e) is not described.

- Figure 7: please indicate the contour of cloud area, since the gray colors marked the area where cloud particles and smoke particles co-exist, as well as the boundaries of clouds.
- Figure 7 caption:  Height-time distributions of the particle backscatter coefficient at 532 nm in the night of 29 - 30 May 2023. Height-time bins with a high fluorescence backscatter coefficient ($> 2.5 \times 10^{-5}$ $Mm^{-1}$ $sr^{-1}$) are colored in gray.
- L406-407: 'in such cases with low but relevant aerosol presence in the cloud surroundings, especially at the cloud top': too long, with lots of ambiguity, therefore not clear. It would be better to put it in a general way:  in such cases where aerosols/smoke particles are identified as INP…

- L421: newly-implemented → newly implemented
- L422: the 2023 summer wildfire season → the summer wildfire season of 2023
- The conclusion part could be re-organized to make it more fluent, compact and direct. Unnecessary sentences like 'back trajectory calculations suggested…' could be distracting (and it has been mentioned before), therefore, reduce the readability.

---

## Author Comment (AC1)

Dear Referee 1!

We thank you for careful reading of the manuscript and the thoughtful questions, comments and suggestions. We have addressed the comments in this reply letter and in the revised version of the manuscript. Most changes are aimed to improve the clarity and presentation of the manuscript.

A version of the revised manuscript with tracked changes is attached to this reply letter. All changes in the revised manuscript are given in BLUE. The old passages that were removed are marked in RED.

In this reply letter, the comments of Reviewer 1 are given in BLACK, our answers are given in GREEN.

**General comment**

This paper demonstrates the detection capability of fluorescence lidar in revealing and identifying high-altitude smoke layers, based on the measurements collected during the Canadian wildfire season in 2023. These layers, due to their low particle concentrations and high altitudes, may be beyond the detection limits of conventional Mie-Raman lidar systems, yet they still influence the ice cloud formation and cloud properties. The authors first introduce the calibration method of the fluorescence channel, followed by an analysis of several scenarios of measurements. These results demonstrate that the fluorescence channel enhances the detection capability of a conventional Mie-Raman lidar, brings new information about aerosol characterization and provides new opportunities for the study of aerosol-cloud interactions. The study aligns well with the scope of <Atmospheric Chemistry and Physics> and the measurements presented are valuable. The presentation of data is clear and the conclusions are scientifically sound. Therefore, I recommend the paper for acceptance. However, major revisions regarding the following aspects are needed:

1. In the calibration procedure, the calculation of detector efficiency contains two terms: the PMT quantum efficiency and the PMT gain ratio. According the text in line 148-153, the quantum efficiency is determined using the data provide by Hamamatsu and the PMT gain ratio is derived by swapping the detectors in the nitrogen Raman and fluorescence channels. However, the description about how this experiment was conducted remains unclear. Please clarify and verify whether the PMT gain ratio calculation has already accounted for the ratio of quantum efficiency.

> ➔ The experiment was conducted as follows: For better clarity we'll name the PMT used in the nitrogen Raman channel as "PMT_R" and the PMT in the fluorescence channel as "PMT_F". First, we measured 5 minutes with the standard configuration (PMT_R in Raman channel and PMT_F in fluorescence channel). Secondly, PMT_R and PMT_F were swapped and again, 5 minutes of measurement were taken. Thirdly, the PMTs were returned to their initial position in the standard configuration. Another 5 minutes of measurement were taken to ensure that the atmospheric conditions stayed rather stable during the time period of the experiment, so that the signals of the consecutive 5-minute periods are comparable. Now, the temporal average of the signals in both channels were calculated for the standard configuration (P_387_0, P_466_0) and the swapped configuration (P_387_swapped, P_466_swapped). For each channel, the ratio of the signals for both configurations was calculated, i.e., P_387_0/P_387_swapped and P_466_swapped/P_466_0. These ratios account for electrical gain only, because each signal ratio is evaluated at a certain spectral range (=

the filter bandwidth). The mean of both ratios was used as PMT gain ratio for fluorescence backscatter calculations.

The ratio of quantum efficiency can't be obtained with this experiment because an absolute reference for the incident signal intensity in the respective detection channel is missing. To obtain the quantum efficiency experimentally, a calibrated lamp with a known spectrum would have to be used. In fact, with such an approach the whole ratio of the channel optical efficiencies could be determined all at once.

2. The writing needs improvements, as some sentence and expressions lack of the precision expected in scientific papers. The paper contains many colloquial phrases that should be avoided to ensure the writing is concise and accurate. Please refer to specific comment for details.

➔ Thank you for your detailed comments and recommendations for improving of the writing.

3. Section 3.2.4 is lengthy and lacks of clear structure, making it difficult to follow. Please consider condensing the section for better readability. The authors could begin by introducing the observations, followed by the analysis that progressively address key aspects.

Thank you for this comment. We rearranged the structure of Section 3.2.4 in order to increase readability and shortened the discussion at some points.

**Specific comments:**

L20: A logic issue with 'By serving as cloud condensation nuclei (CCN) or ice nucleating particles (INPs)…, the microphysical properties of water clouds…

➔ To avoid confusion, the "water" was removed, now generally speaking of clouds. References were added to show that there is also an aerosol effect on ice clouds similar as the Twomey effect for liquid water clouds.

L41: "Laser-induced fluorescence is a known process and several remote-sensing applications are based on it." ➔ Laser-induced fluorescence is a well-established technique, serving as the basis of several remote sensing applications.

➔ Change has been applied

L43: In the atmosphere ➔ In the domain of atmospheric research

➔ Change has been applied

L51-52: "but most were small…" do you mean these lidars were with simple configuration? And please add reference.

➔ Yes, it was meant, that these lidars described by Rao et al. (2018) and Li et al. (2019) were with simpler configuration.
➔ Also due to a comment of the other reviewer, the whole paragraph was rearranged, so that this statement was not necessary anymore.

L111: "and only a little loss of fluorescence return" ➔ "with minimal loss of the fluorescence return"

➔ Change has been applied

L114: only if no rain is expected ➔ only when no rain is expected

➔ Change has been applied

L114: Complete nights were collected since 2022 … ➔ Complete night measurements have been collected since 2022…

➔ Change has been applied

L116: the derivation of  new products

➔ Change has been applied

L117: similarly as ➔ similarly to

➔ The phrase has been changed as follows: "similar to Veselovskii et al. (2020)".

L119, L120: The transmission at elastic wavelenght is missing. It is important for the validity of lidar equations, although the 2 terms cancel out in Equation 3.

➔ This is correct, of course. The atmospheric transmission at the elastic wavelength was added to Eqs. (1) and (2) as $T_L$ (transmission at the laser wavelength)

L122: "and the CR and CF represent the corresponding lidar calibration constants."

➔ Change has been applied

L210: …polluted troposphere, with overall aerosol optical depth (AOD) of around 0.8 at 532 nm.

➔ Change has been applied

L212: …and 17 UTC on 04 July… ➔ …**at 17:00 UTC** on 04 July 2023. Please use standard time notation and keep it consistent everywhere in the manuscript.

➔ We revised the notation of times and dates to keep it consistent throughout the whole manuscript.

Figure 2: what are the white points in lidar quicklooks?

➔ On 4-5 July 2023 there were optically thick clouds present. Thus, we have no reliable lidar data above (low and/or negative signals in elastic channels). This leads to a very noisy appearance (high values) of the fluorescence backscatter coefficient. Therefore, these data points were filtered by the following criterion for the range-corrected signal at 1064 nm: $P_{1064} < 5 \times 10^4$. Thus, also some noisy points in the cloud-free periods were excluded. This led to some white points in the time-height plots.

Figure 2(d) is too crowded and the profile of fluorescence backscatter coefficient is truncated, please split this figure into 2, for clarity and better presentation.

➔ Thank you for the comment. We revised the figure and split the vertical profiles into two subfigures.

L217-220: Please indicate the vertical range which you are describing. Such lidar ratios were detected in tropospheric layer or UTLS layer?

➔ The aerosol layer described here is the tropospheric layer from 3.4 to 5.8 km height. The corresponding height information has been added in the manuscript as well.

L243: 22 UTC…. indicated aerosol presence ➔ 22:00 UTC…were presented in the atmosphere. Just for logic consistency, since a cloud layer also existed.

➔ The sentence was changed to:
"Only the polluted boundary layer and some thin layers up to 4 km height indicated aerosol presence, and a thin cloud was visible at around 4 km height from 21:00 to 22:00 UTC."

L245: This already illustrates  that with measurements…

➔ Change has been applied

L252: the 532 nm backscatter coefficient is  only slightly enhanced

➔ Change has been applied

Figure 3: the curves are too crowded, if you split them into 2: elastic backscattering + fluorescence backscatter with capacity, it will look better. And the peak at about 2.2 km seems to be cloud at around 19:10 UTC.

➔ Thank you for the comment. We revised the figure and split the vertical profiles into two subfigures. Furthermore, we chose a smaller height range for the vertical profiles to present the relevant height ranges clearer.

L251-257: It seems, in this case, the detection ability of elastic channels is already at their limit. The description about maxima is not quite evident according to the figure.
How did you smooth the profiles? Is the smoothing method wavelength dependent or vertically varying? How come the backscattering at 532 fails to resolve the aerosol layer below 6.5 km but is still able to resolve the maxima at above 9 km?
This case shows the detection limit of 532 nm backcattering is about 0.2 Mm-1sr-1, is it fair to say so? Do you have any comment about the vertical variation of the fluorescence capacity? Are they real and link to smoke properties/compositions or more likely to be artifacts produced by the vertical variation of elastic and/or fluorescencence backscattering and/or smoothing method?

➔ Thank you for your two comments concerning the presentation of Figure 3. It was reshaped to make it clearer and that the maxima described in the text are better recognizable in the new (sub-)figures.
➔ The description of the maxima at 6.5 km was changed to the following:
"For the layer at 6.5 km height, their corresponding maxima are at higher altitudes than the distinct maximum in the fluorescence backscatter coefficient. The 355 nm backscatter coefficient even fails to resolve the aerosol layer at 6.5 km. However, all

elastic-backscatter detection channels reach their limits with the two high layers at 9 and 9.75 km altitude."

➔ Yes, the smoothing is done wavelength-dependent. Generally, 355 nm and 1064 nm profiles are smoothed more strongly than the one at 532 nm. The fluorescence backscatter is always smoothed in the same way as the 532 nm backscatter. In this particular case, 532 and 1064 nm are smoothed in the same way. 355 is smoothed a bit stronger. However, we tested it with the same smoothing for all 3 wavelengths. But this did not improve the 355 nm backscatter ability to resolve the layer maxima.

➔ Regarding the resolution of the aerosol layers: To resolve these layers is generally difficult, as they are optically thin. But in principle, $\beta_{532}$ resolves all the layers, even if sometimes only with a broad increase rather than a distinct peak.
Regarding your particular comment to the layer at around 6.5 km: Probably you refer to the fact that for this layer, the maxima of the 532 nm and 1064 nm backscatter coefficients are at slightly higher altitudes than the distinct peak in the fluorescence backscatter coefficient (this remark was also added in the manuscript). Here is an attempt to explain: Both elastic backscatter coefficients show an increase from around 6 km on, where also the fluorescence backscatter coefficient starts to increase. While $\beta_F$ already peaks at 6.4 km, $\beta_{532}$ reaches its maximum at 6.46 km and $\beta_{1064}$ only at 6.8 km. Above their maxima, $\beta_{532}$ and $\beta_{1064}$ stay enhanced compared to the background up to an altitude of 7.7 km and so does $\beta_F$ as well, but at clearly lower values than at the peak at 6.4 km. In combination, this sequence of the backscatter profiles could indicate the presence and increasing influence of less fluorescing aerosol particles from 6.5 to 7.7 km height. So, the different altitudes of the maxima could be due to changing aerosol characteristics at this altitude range. But this is just a hypothesis.

➔ In summary, we would argue that also this layer at 6.5 km is resolved by $\beta_{532}$ due to its broad increase at this altitude range and the shift in the maximum may be due to the presence of less fluorescent aerosol.

➔ **Detection limit:** For the layer at 3.3 km, which is still clearly resolved, $\beta_{532}$ peaks at $0.15\ \mathrm{Mm^{-1}\ sr^{-1}}$, the higher layers range around $0.1\ \mathrm{Mm^{-1}\ sr^{-1}}$. Thus, we would assume a detection limit of the 532 nm backscatter coefficient of $0.1 - 0.15\ \mathrm{Mm^{-1}\ sr^{-1}}$. These values also agree well with the two other case studies (15 May and 29 May 2023).

➔ **Vertical variation of the fluorescence capacity:** As $\beta_F$ and $\beta_{532}$ are smoothed in the same way, smoothing effects on the fluorescence capacity can be ruled out.
In this particular case, the fluorescence capacity is difficult to evaluate as the aerosol layers are optically very thin and the elastic backscatter values are close to the background noise level and can hardly be analyzed. Thus, in this case, the fluorescence backscatter coefficient is the more interesting quantity. In the other cases presented in this manuscript with optically thicker aerosol layers, the variations in fluorescence capacity should link to the aerosol properties. For further details, please refer to our reply to your corresponding comment to the case study of 15 May 2023.

L258: (also because the particle depolarization ratio is also quite low with 2 %, not shown) ➔ especially because the particle depolarization ratio (not shown) is also quite low, around 2 %.

➔ Change has been applied

L269: (again in agreement with AERONET: 0.1 at 500 nm) ➔ which is consistent with AERONET data showing 0.1 at 500 nm

➜ Change has been applied

L264: Another example of such "unnoticeable" layers is the night of  15-16 May 2023. Similar error in L266.

➜ Changes have been applied (also at a few other places in the manuscript)

L268: In the range of 4-6.7 km, the vertical variation of fluorescence capacity is significant, from 2*10-4 to 6*10-4, such strong variation is also observed in the layer at 10.5-12.2 km. The comparability of fluorescence capacity between upper layer and lower layer depends on the selection of vertical range. So, again, the question is: do you think this variation of fluorescence capacity is real? Why would it change so much in the same plume? Is there any explanation for the drop of fluorescence capacity at the edges of the plume? Or it is just some artifacts arising from different sensitivity of detection channel or smoothing ?

➜ In this case, the variations in fluorescence capacity should be real and not systematic. Such problems with artifacts from low signal intensity may only be relevant in the case of optically very thin aerosol layers, as it was the case on 21 September 2022, where the elastic backscatter coefficients were so small that they can hardly be analyzed. Please refer to the corresponding comment above. Smoothing effects have already been ruled out there as well.

➜ At closer inspection, the "layer" in the range 4-6.7 km seems to be composed of 3 individual layers. This impression is given by 3 individual peaks in the vertical profiles of $\beta_F$ and $G_F$ in Fig. 4(d) and is corroborated by the temporal evolution before the averaging period, which can be seen in the time-height plot of the fluorescence backscatter coefficient in Fig. 4(b). Between these 3 layers, slight variations in the fluorescence capacity can be expected. And when we compare the maxima of these 3 layers, the fluorescence capacity varies from $5.1 \times 10^{-4}$ (4.5 km) to $6 \times 10^{-4}$ (5.0 km), which is in our opinion a quite reasonable variation for slightly different smoke layers.

The main variations that you are describing are found at the edges of the smoke plumes. This dropping at the plume edges, which you also mentioned in your comment, could have different causes: Firstly, the number of fluorescing particles could decrease towards the edges of the plume, as the highest concentration of smoke particles is expected in the plume center. Thus, the relative influence of other less or non-fluorescing background aerosol particles could increase, reducing the fluorescence capacity. This may especially be important, in cases where a layer of less or non-fluorescing aerosol particles is present above and below and some mixing occurs at the edges of both plumes. Secondly, and probably more important in this case, is the height constancy of the fluorescing aerosol layer. If the height of the layer varies during the averaging period, data points outside the plume (where the fluorescence capacity is of course lower) are averaged together with data points inside the plume. As a result, the temporal mean value of the capacity at the plume edges is lower than in the center of the plume, where in-plume data points are averaged over the entire averaging period.

This effect can be nicely observed in Fig. 2 for the measurement case on 4 July 2023. The thick smoke layer from 3.4 to 5.8 km height is rather homogeneous and its lower edge is very height-constant throughout the whole averaging period from 21:00 to 22:00 UTC. As a result, the increase in fluorescence capacity at the base of the smoke layer is very steep (cf. Fig. 2d). In contrast, the layer top height clearly decreases

during the averaging period, so that the decrease in fluorescence capacity is much flatter at the top than the increase at the layer base.

L269,270: What is the travel time of this smoke layer, at 4-6.7 km? And how many days does it need to become so spherical? What is the relative humidity in this layer? The elastic backscatter coefficeint is so low in this layer that there appears to be no evidence of hygroscopic growth. Does lidar ratio tell something?

➔ The calculated back trajectory suggests a transport time of approximately 10 days for layer 1 (cf. Fig. A). The relative humidity in layer 1 was in the range of 35-45% according to our lidar data.

➔ The hypothesis of hygroscopic growth influencing the fluorescence capacity of layer 1 was removed due to the lack of evidence and relevance for the main message of this manuscript. Therefore, we restricted to the analysis of the aerosol optical properties (fluorescence, depolarization ratio, lidar ratio and Angström exponent) and naming possible causes for the differences.

L275: Does the back trajectory reveal any difference in transport time or pathway between the two layers?

➔ The back trajectory analysis for Leipzig at the starting time of 2 UTC is displayed in Fig. A. The main difference lies in the altitude of the back trajectories. Layer 2 seems to be transported constantly at around 10-11 km height (which was at least at our observation place and time slightly above the tropopause). Instead, layer 2 mainly ranged at altitudes from 5-6 km, for a short time reaching 8 km height at the maximum.

➔ Also, the back trajectories suggest a shorter travel time of layer 2 (5-6 days) compared to approximately 10 days of layer 1.

➔ The shorter travel time and lower humidity (because of higher altitude, shortly above the tropopause) indicate less aging of layer 2 compared to layer 1.

[Figure]

**Figure A: Back trajectory analysis for Leipzig at 02:00 UTC on 16 May 2023.**

L276: 'the aforementional Canadian wildfires' : not clear which wildfire is it referring to

➔ These simple back trajectory analysis does not allow the attribution to a certain wildfire source. Only a general source region can be identified. In general, the entire paragraph has been revised and the sentence mentioned in your comment has been omitted.

L288: '…reveals another aerosol layer": this layer is also called "another ..layer" in L271. Better to name the layers to be 'Layer 1, 2,…' in each case to improve the clarity.

➔ Thank you for the hint. The layers have been numbered accordingly.

L307-309: This long sentence should better be rephrased to improve clarity.

➔ Indeed. The sentence and the whole paragraph have been reformulated to improve clarity and accuracy.

L310: Is it range-corrected signal in Figure 5(b)? Is the fluorescence signal rescaled? What is the definition of the unit 'MC*s-1'? Are they from analog channels or photon-counting channels?

➔ Yes, Fig. 5(b) showed the range-corrected signal. To improve the accuracy of the explanations, this plot was replaced by the background-corrected signal, which units are megahertz (MHz).

L315: only a part of the actual (strong) backscatter signal: (strong) ? this is confusing.

➔ This part has been rephrased. Thanks for the remark.

L316,317: please provide the reference of the overlap correction. Are the profiles shown in previous figures corrected from overlap? Why are they cut below 1 km?

➔ We computed backscatter coefficients from the ratio between the elastic and Raman channel, which in principle does not suffer from overlap effects. Because our system has a large telescope and narrow field-of-view, the full overlap is reached only at about 2.5 km. We can extend the observational capabilities to the ground but in the first kilometer some interchannel-overlap and deadtime effects may introduce artifacts into the retrieved aerosol information. For this reason, we cut the signal at 1.2 km routinely.
The overlap function is based on Raytracing calculations. The correction is relevant for the extinction coefficient. The study focuses rather on the backscatter coefficient. This aspect has been removed from the paragraph, to keep the description as clear and accurate as possible focusing more in the sensitivity of the system.

L324: dynamical ➔ dynamic

➔ The word has been corrected. Thanks.

L324-325: …much smaller, and because of that, the… ➔ …much smaller. Consequently, the…

➔ Thank you for the hint. The sentence has been modified.

L332: 'UTLS' defined twice

➔ Thank you for the hint. The second definition was removed.

Figure 6: please add the profiles of temperature and relative humidity with respect to ice.

➔ The profiles of temperature and relative humidity obtained from radiosonde measurements at Lindenberg were added as Fig. 6(e).

L346: "water content can quench fluorescence scattering": this sentence is not clear enough. How does water content influence fluorescence capacity, by suppressing fluorescence emission or enhancing elastic scattering?

➔ As also explained in the reply to the other reviewer: To mention quenching here, makes no sense. The low fluorescence capacity of cloud particles is due to their strong elastic backscattering, while their fluorescence backscattering is low, as pure water does not fluoresce. The corresponding part of the sentence has therefore been corrected. For further details, please refer to the reply to the other reviewer.

L349: separate → differentiate

→ Change has been applied

L352,353: 'Furthermore, the smoke layer rose…the cloud top rose first, and later, become scattered and seemed to dissolve. All these facts indicate that the smoke particles may have triggered the cloud formation by serving as INPs'. The link between smoke particles serving as INPs and the movements of clouds and smoke layers is not clearly explained here. In addition, as clouds became thinner and thinner, the smoke layers seemed to get thicker, is it related to ice crystal formation?

→ The link between the temporal evolution of the smoke layer and the cirrus clouds could be that smoke particles are serving as INPs. The important observation is here, that the presence of cirrus clouds coincides temporally and spatially with the presence of smoke particles. From the time at which the smoke layer was located exclusively at higher altitudes than the altitude at which the cloud formation took place, ice nucleation appears to cease and cirrus clouds no longer form. If the ice nucleation would not be related to the smoke particles but to background aerosol only, one would not expect this stop in ice nucleation and cloud evolution. Therefore, this observation suggests that the smoke particles act as INPs in this case.

→ To comment on the thickening of the smoke layer over time would be rather hypothetical. Of course, looking at the lower fluorescence backscatter coefficient at the time, where the smoke layer seemed to be partly inside the cloud (before 23:30 UTC), one could speculate that ice nucleation and subsequent falling of the ice crystals may have reduced the optical thickness of the smoke layer at that time. However, it could also simply be a transport-related thickening of the smoke layer. So, the answer to your question is: We can't tell…

L354: "the clouds even became scattered and seemed to dissolve": why do ice clouds dissolve at such temperature?

→ Thank you for your comment. You are right, "dissolve" is probably the wrong vocabulary here. As we are monitoring with lidar the temporal evolution of the atmospheric column over a fixed location, the absence of cirrus clouds at the later time steps does not mean that the cirrus clouds observed at previous time steps did really dissolve, but that no cloud formation happened anymore at the later time steps. So, the following formulation is probably better: "The clouds even became scattered and the ice nucleation and cloud formation seemed to stop."

→ The change was applied to the manuscript.

L369: on 29 May → in the night of 29 May

→ Change has been applied

L371: 21 to 22 UTC → 21:00 to 22:00 UTC

→ Change has been applied

L371: please indicate the two nucleation sections in Figure 6(d)

→ Horizontal dashed lines were added to Fig. 6(d) to indicate the nucleation sections.

L375: 'Above, the high fluorescence capacity…': In the context, 'Above' does not specify what you are referring to.

➔ It was changed to "At around 12 km height, …"

L376-382: A general comment regarding the description of figures: when presenting values in a vertical profile, please include their corresponding vertical levels rather than relying only on descriptive phrases like 'down to the upper boundary of the lower cloud part' or 'at the top of the lower part of the cirrus clouds'…

➔ The vertical levels were either added or used to replace the descriptive phrases.

L382-383: If this difference in fluorescence backscattering can be explained by fluorescence quenching, then why did it occur specifically at the cloud top of the upper cirrus part, rather than elsewhere?

➔ First of all, fluorescence quenching is only one possibility. The height level of the reduction in the fluorescence backscatter coefficient seems to coincide with the upper nucleation zone, where ice crystals are forming. But whether this would increase fluorescence quenching is hypothetical.
➔ However, the removal of smoke particles (that served as INP) from this altitude due to the falling of ice crystals seems to be a more plausible explanation as it is also stated in the revised version of the manuscript.

L390: 'ranged from (at) the minimal values of…to…'

➔ We meant that the values of $10^{-6}$ to $10^{-5}$ are minimal compared to other regions and cases. We replaced minimal by "very low" to make this clearer.

L392--394: ' In this case,…again up to 1.4*10-5Mm-1sr-1 ." --- the fluroescent layer at around 7 km seemed not in contact with cirrus clouds after 21:00, and there was no fluorescence between 7.5 km and 8.5 km. Therefore, I am not convinced that this layer was due to the accumulation of smoke particles at cloud base.

➔ We disagree in this point. In our opinion, the missing contact of the fluorescing layer to the cirrus cloud does not argue against the hypothesis of accumulation of smoke particles at the cloud base. With lidar measurements, we only get a "snapshot" at a certain time and location. We do not know what happened to a certain air mass some minutes before we observed it with the lidar. The sublimation of the ice crystals may have happened some time before, so that the remaining smoke particles were later observed a bit below the cloud base. Furthermore, the vertical profile of the relative humidity over ice from the radiosonde at Lindenberg in Fig. 6(e) supports our hypothesis of scavenging and subsequent accumulation of smoke due to sublimation at the cloud base. Around 8 km altitude, RH_ice drops below 100% and decreases to values of 50-60% at around 7 km. In such humidity conditions, falling ice crystals will sublimate, leaving only remaining smoke particles, that had served as their INPs before. And this is also supported by our lidar observations. The fluorescence capacity plot in Fig. 6(c) reveals such strongly fluorescing layers not only at 7 km height between 21:00 and 22:00 UTC, but also between 01:15 UTC and 02:00 UTC at around 8 km height.

Figure 6(e) is not described.

➔ Figure 6(e) was replaced by a figure showing temperature and relative humidity over ice from a radiosonde at Lindenberg at 18 UTC on 29 May 2023.

Figure 7: please indicate the contour of cloud area, since the gray colors marked the area where cloud particles and smoke particles co-exist, as well as the boundaries of clouds.

➔ The top of the cirrus cloud layer has been marked in black in Fig. 7.

Figure 7 caption: Height-time distributions of the particle backscatter coefficient at 532 nm in the night of 29 - 30 May 2023. Height-time bins with a high fluorescence backscatter coefficient (> 2.5 ×10−5 Mm−1 sr−1) are colored in gray.

➔ Change has been applied

L406-407: 'in such cases with low but relevant aerosol presence in the cloud surroundings, especially at the cloud top': too long, with lots of ambiguity, therefore not clear. It would be better to put it in a general way: in such cases where aerosols/smoke particles are identified as INP…

➔ The important point in this statement is that the fluorescence lidar may be used to provide INP information also in cases with **low aerosol load**. Therefore, this information cannot be omitted here.

L421: newly-implemented ➔ newly implemented

➔ Change has been applied

L422: the 2023 summer wildfire season ➔ the summer wildfire season of 2023

➔ Change has been applied

The conclusion part could be re-organized to make it more fluent, compact and direct. Unnecessary sentences like 'back trajectory calculations suggested…' could be distracting (and it has been mentioned before), therefore, reduce the readability.

➔ The statement regarding the back trajectory calculations was removed.

[revised manuscript text omitted]

$$G_\mathrm{F}^{355} = \frac{\beta_\mathrm{F}}{\beta_{355}\, d_\mathrm{IF}}, \tag{7}$$

where $d_\mathrm{IF} = 44\,\mathrm{nm}$ is the bandwidth of the interference filter in the fluorescence channel. Furthermore, data from the collocated portable Raman lidar Polly$^\mathrm{XT}$ (Engelmann et al., 2016) at TROPOS were used to provide quality-assured depolarization profiles.

**3 Observational results**

Due to the broad bandwidth of the fluorescence channel and the low intensity of the fluorescence signal, measurements were only possible during the night. At daytime, scattered solar radiation would cause too much noise in the fluorescence channel. As the MARTHA system is operated manually, the number of measurements remains limited. Since August 2022, about 50 measurements have been performed, providing more than 250 hours of atmospheric fluorescence observations. Typical atmospheric values of the fluorescence backscatter coefficient and fluorescence capacity, that were obtained at Leipzig during the time period from August 2022 to October 2023, are presented in the next paragraph.

In general, $\beta_\mathrm{F}$ ranged between $1\times10^{-5}\,\mathrm{Mm^{-1}\,sr^{-1}}$ for background aerosol and more than $1\times10^{-3}\,\mathrm{Mm^{-1}\,sr^{-1}}$ for optically extraordinarily thick wildfire smoke layers. Correspondingly, $G_\mathrm{F}$  ($(G_\mathrm{F}^{355})$ varied from $\sim10^{-6}$  ($\sim10^{-8}\,\mathrm{nm^{-1}}$) for clouds and $1\times10^{-5}$ ($1\times10^{-7}\,\mathrm{nm^{-1}}$) for background aerosol to $1.3\times10^{-3}$ ($1.3\times10^{-5}\,\
[revised manuscript text omitted]